
# Towards kilometer-scale ocean–atmosphere–wave coupled forecast: a case study on a Mediterranean heavy precipitation event

César Sauvage[1,a], Cindy Lebeaupin Brossier[1], and Marie-Noëlle Bouin[1,2]

[1]CNRM, Université de Toulouse, Météo-France, CNRS, Toulouse, France
[2]Laboratoire d'Océanographie Physique et Spatiale, Ifremer, University of Brest, CNRS, IRD, Brest, France
[a]now at: Physical Oceanography Department, Woods Hole Oceanographic Institution, Woods Hole, MA, USA

**Correspondence:** César Sauvage (sauvagecesar@hotmail.fr)

**Abstract.** The Western Mediterranean Sea area is frequently affected in autumn by heavy precipitation events (HPEs). These severe meteorological episodes, characterized by strong offshore low-level winds and heavy rain in a short period of time, can lead to severe flooding and wave-submersion events. This study aims to progress towards integrated short-range forecast system via coupled modelling for a better representation of the processes at the air–sea interface. In order to identify and quantify the

coupling impacts, coupled ocean–atmosphere–wave simulations were performed for a HPE that occurred between October 12 and 14, 2016 in the South of France, using the coupled AROME-NEMO-WaveWatchIII system and notably compared to atmosphere-only, coupled atmosphere–wave and ocean–atmosphere simulations. The results showed that the HPE fine-scale forecast is sensitive to both couplings: The interactive coupling with the ocean leads to significant changes in the heat and moisture supply of the HPE that intensify the convective systems, while coupling with a wave model mainly leads to changes

in the low-level dynamics, affecting the location of the convergence that triggers convection over sea. Even if this first case study with the AROME-NEMO-WaveWatchIII system does not clearly show major changes in the forecasts with coupling and highlights some attention points to follow (ocean initialisation notably), it illustrates the higher realism and potential benefits of kilometer-scale coupled numerical weather prediction systems, in particular in case of severe weather events over sea and/or in coastal areas, and shows their affordability to confidently progress towards operational coupled forecasts.





## 1 Introduction

In the last decade, improving the forecast of intense weather events involving air–sea interactions has motivated operational forecast centres to develop and operate ocean–atmosphere–waves coupled modelling platforms for short- and medium-range weather predictions [see for instance the Geophysical Fluid Dynamics Laboratory (GFDL) model used at the National Weather Service, Bender et al. (2007); the Coupled Ocean/Atmosphere Mesoscale Prediction System for Tropical Cyclones (COAMPS-

TC) operated at Naval Research Laboratory for hurricane prediction, Doyle et al. (2014); the global ocean–ice–atmosphere coupled prediction system run at Environment and Climate Change Canada, Smith et al. (2018); and the recent developments at the European Centre for Medium-Range Weather Forecasts, Magnusson et al. (2019)].

Tropical cyclones (TCs) above all have been known for long to be impacted by the surface cooling of the ocean they generate (e.g. Bender et al., 1993; Bender and Ginis, 2000; Bao et al., 2000). Realistic simulations have shown that the initial state of

the ocean, namely the sea surface temperature (SST) and stratification may significantly reduce the TC intensity (e.g. Chan et al., 2001). Several large-scale studies have shown that using ocean–atmosphere coupling improves in a statistical way the prediction of TCs with respect to atmosphere-only simulations in every cyclonic basin (e.g. Bender et al., 2007; Samson et al., 2014; Mogensen et al., 2017; Lengaigne et al., 2018). Using 3D ocean models in coupled configurations is mandatory to accurately represent the complex subsurface processes (e.g. upwelling) responsible of the SST cooling (Yablonsky and Ginis,

2009). As TC development is known to be sensitive to both enthalpy and momentum transfer coefficients (Emanuel, 1986), taking into account the wave impact on the sea surface roughness can also influence the TC representation in numerical models. Case studies using ocean–atmosphere–waves coupled configurations showed an influence of wave growth on the TC intensity and development (e.g. Olabarrieta et al., 2012; Lee and Chen, 2012; Doyle et al., 2014; Pianezze et al., 2018). Sensitivity tests using representation of the surface fluxes including the impact of sea spray showed more contrasted results, depending on

the parameterization used and on the case studied (e.g. Wang et al., 2001; Gall et al., 2008; Green and Zhang, 2013; Zweers et al., 2015). Most of the coupled configurations used for improving the TC forecast have horizontal resolutions of 10–25 km, enabling them to cover large oceanic basins and fine enough to properly represent relatively large-scale events like TCs. Only recent case studies make use of kilometric horizontal resolutions permitting to simulate more accurately the fine-scale processes within the TC structure (e.g. Lee and Chen, 2012; Green and Zhang, 2013; Pianezze et al., 2018).

Extremes events also often occur in the Mediterranean Sea. For instance, medicanes are severe storms looking like TCs in their developed phase, although smaller in size and weaker (e.g. Lionello et al., 2003; Renault et al., 2012; Ricchi et al., 2017; Varlas et al., 2018; Bouin and Lebeaupin Brossier, 2020b; Varlas et al., 2020). In medicanes as in tropical cyclones, ocean surface cooling is observed, primarily affecting the heat and moisture exchanges. Case studies based on coupled simulations gave contrasting results on the impact of the feedback from the waves or the ocean on medicanes. For instance, Ricchi et al.

(2017) investigating the medicane of November 2011 using COAWST (Coupled Ocean Atmosphere–Wave Sediment Transport, Warner et al., 2010) at 5 km resolution and Bouin and Lebeaupin Brossier (2020b) studying the one occurring in November 2014 through high-resolution coupling [1.3 km for the atmosphere using MESO-NH (Mesoscale Non-Hydrostatic Model - Lac et al., 2018) and 1/36° for the ocean using NEMO (Nucleus for European Modelling of the Ocean - Madec and the





NEMO system team, 2008)] showed that the direct impact of the ocean coupling did not significantly change the track and
intensity of the medicanes. Ricchi et al. (2017) suggested nevertheless that the way to calculate the sea surface roughness, and
more generally the air–sea processes, can affect significantly the results by notably playing on the intensification of the near
surface wind. Also, Varlas et al. (2020) showed an overall improvement of the forecast skill over sea using a two-way coupling
between the atmosphere and waves, respectively WRF (Weather Research Forecasting - Skamarock et al., 2008) and WAM
(the ocean WAve Model - The Wamdi Group, 1988) models.

Generally related to cyclogenesis, the Mediterranean Sea is also prone to high and local wind from continental origin, chan-
nelled and accelerated in the steep surrounding valleys, such as Mistral or Bora, which last usually several days and generate
very rough sea states, and sometimes result in strong damages (e.g. Ardhuin et al., 2007). Several case studies investigated the
impact of Mistral or Bora wind on the ocean and the impact of using ocean–atmosphere, or atmosphere–waves coupled models
(e.g. Loglisci et al., 2004; Pullen et al., 2007; Small et al., 2012; Ricchi et al., 2016; Ličer et al., 2016; Seyfried et al., 2019).
They showed a quick evolution of the SST and currents during this type of events, with a significant feedback on the surface
heat and momentum fluxes, but no significant change on the low-level atmospheric flow.

In the present study, we investigate the impact of ocean–atmosphere–wave coupling on a different kind of Mediterranean
extreme weather event, namely a heavy precipitation event (HPE, Ducrocq et al., 2014, 2016). Such events generally occur
in autumn and are characterized by large amount of precipitation over a small area in a very short time, causing huge flash
floods leading to considerable damages and numerous casualties (e.g. Petrucci et al., 2019). These events are usually generated
by quasi-stationary mesoscale convective systems (MCSs) fed by strong offshore low-level winds over a warm Mediterranean
Sea. Air–sea processes are thus key elements in the development of those HPEs (e.g. Duffourg and Ducrocq, 2011). Rainaud
et al. (2017) using the coupling between the WMED configurations of AROME (Application of Research to Operations at
MEsoscale - Seity et al., 2011; Fourrié et al., 2015) atmosphere model at 2.5 km resolution and NEMO at a 1/36°-resolution
(Lebeaupin Brossier et al., 2014), re-assert the importance of an interactive ocean and its impact on the surface evaporation
water supply for HPEs. In addition to this, Thévenot et al. (2016); Bouin et al. (2017); Sauvage et al. (2020) showed the im-
portance of taking the sea state into account in the calculation of air–sea fluxes during Mediterranean HPEs, with a significant
impact on the location of the heavy precipitation. Indeed, the parameterization of sea surface turbulent fluxes is key in rep-
resenting the exchanges between the different compartments. Generally implemented as bulk parameterizations (e.g. Coupled
Ocean–Atmosphere Response Experiment (COARE) 3.0, Fairall et al., 2003), several formulations enable to represent the sea-
state impact on the momentum and heat fluxes (Oost et al., 2002; Taylor and Yelland, 2001; Sauvage et al., 2020).

The studies listed above demonstrate the interest of more complete regional simulating systems to better predict high-impact
events involving air–sea interactions, and of combining the capabilities of fine-scale (1 to 2 km in horizontal resolution) models
with ocean–atmosphere–waves coupling. Also, the continuous increase in high-performance computing capabilities fosters the
development of such coupled modelling systems with kilometric resolution and makes them usable for operational forecasting
(e.g. Pullen et al., 2017; Lewis et al., 2018, 2019a,b,c).





In this context, the present study describes a new kilometric regional coupled system involving the Météo-France high-resolution operational numerical weather prediction (NWP) model AROME-France, the WaveWatch III wave model (hereafter WW3, Tolman, 1992) and the NEMO ocean model, which paves the way to the future coupled regional convection-resolving NWP system of Météo-France. This system will be used here to assess the coupling impacts during an HPE which occurred from 12 to 14 October 2016.

A detailed description of the coupled system is given in Sect. 2. The main characteristics of the studied HPE and the numerical set-up are presented in Sect. 3. Then the contribution of the two-way coupled atmosphere–wave and atmosphere–ocean is analysed in Sect. 4. In Sect. 5 the results obtained using the ocean–atmosphere–wave system are discussed. Finally, conclusions are given in Sect. 6.

## 2 The ocean–atmosphere–wave coupled system

This section presents the tri-coupled system, that combines the ocean-atmosphere coupling previously developed between AROME and NEMO by Rainaud et al. (2017) and the wave-atmosphere interactive exchanges with the AROME-WW3 coupling as fully described by Sauvage et al. (2020). The details of the models configurations and the exchanges management are given in the following for clarity purpose.

### 2.1 The component models

#### 2.1.1 The atmospheric model

The non-hydrostatic AROME NWP model is used in this study, with the same forecast configuration as the one operationally used at Météo-France in 2016 (AROME-France, cy41t1, Seity et al., 2011; Brousseau et al., 2016) with a 1.3 km horizontal resolution and a domain centred over France (Fig. 1), which notably covers the north-western Mediterranean Sea. The vertical grid has 90 hybrid $\eta$-levels with a first-level thickness of almost 5 m. The time step is 50 s.

In AROME, the advection scheme is semi-Lagrangian and the temporal scheme is semi-implicit. The 1.5-order turbulent kinetic energy scheme from Cuxart et al. (2000) is used. Due to its high resolution, the deep convection is explicitly solved in AROME, whereas the shallow convection is solved with the eddy diffusivity Kain–Fritsch (EDKF, Kain and Fritsch, 1990) parameterization. The ICE3 one-moment microphysical scheme (Pinty and Jabouille, 1998) is used to compute the evolution of five hydrometeor species (rain, snow, graupel, cloud ice and cloud liquid water). Radiative fluxes are computed with the Fouquart and Bonnel (1980) scheme for short-wave radiation and RRTM (Rapid Radiative Transfer Model, Mlawer et al., 1997) scheme for long-wave radiation. The surface exchanges are computed by the SURFace EXternalisé (SURFEX) surface model (Masson et al., 2013) considering four different surface types: land, towns, sea and inland waters (lakes and rivers). Output fluxes are weight-averaged inside each grid box according to the fraction of each respective tile, before being provided to the atmospheric model at every time step. Exchanges over land are computed using the ISBA (Interactions between Soil, Biosphere and Atmosphere) parameterization (Noilhan and Planton, 1989). The formulation from Charnock (1955) is used for





inland waters, whereas the Town Energy Balance (TEB) scheme is activated over urban surfaces (Masson, 2000). The treatment
of the sea surface exchanges in AROME-SURFEX is done here with the WASP (Wave-Age-dependent Stress Parameterization)
scheme, detailed in Sauvage et al. (2020) and below; and the albedo is computed following the Taylor et al. (1996) scheme.

### 2.1.2 The ocean model

The NWMED72 configuration of the NEMO ocean model (version 3_6; Madec and the NEMO team, 2016) presented in
(Sauvage et al., 2018) is used here. It covers the north-western Mediterranean basin (Fig. 1), with a 1/72° horizontal resolution
(from 1 to 1.3 km resolution) and uses 50 stretched $z$-levels in the vertical, with a first-level thickness of 0.5 m. This configura-
tion has 2 two open boundaries: a south open boundary near 38°N south of the Balearic Islands and Sardinia, and an east open
boundary across the Tyrrhenian Sea (12.5°E).

In NWMED72, the Total Variance Dissipation (TVD) scheme is used for tracer advection in order to conserve energy and
enstrophy (Barnier et al., 2006). The vertical diffusion follows the standard turbulent kinetic energy formulation of NEMO
(Blanke and Delecluse, 1993). In case of unstable conditions, a higher diffusivity coefficient of 10 m$^2$ s$^{-1}$ is applied (Lazar
et al., 1999). The sea-surface height is a prognostic variable solved thanks to the filtered free-surface scheme of Roullet and
Madec (2000). A no-slip lateral boundary condition is applied and the bottom friction is parameterized by a quadratic function
with a coefficient depending on the 2D mean tidal energy (Lyard et al., 2006; Beuvier et al., 2012). The diffusion is applied
along isoneutral surfaces for the tracers using a Laplacian operator with the horizontal eddy diffusivity value $\nu_h$ fixed at
15 m$^2$ s$^{-1}$. For the dynamics (velocity), a bi-Laplacian operator is used with the horizontal viscosity coefficient $\eta_h$ fixed at
$1.10^8$ m$^4$ s$^{-1}$. The time step is 120 s.

The runoff forcing consists in daily observations for 25 French rivers around the north-western Mediterranean Sea (see
Sauvage et al., 2018, for the complete list) collected from the *banque hydro* database (hydro.eaufrance.fr), and in the monthly
climatology of Ludwig et al. (2009) for Ebro, Júcar and Tiber rivers temporally interpolated to give daily values. Each river
inflow is injected in one grid point in surface (as precipitation).

### 2.1.3 The wave model

The wave model is WW3 (Tolman, 1992) in version 5.16 (The WAVEWATCH III Development Group, 2016). The WW3
domain and bathymetry correspond to the NEMO-NWMED72 grid (at a 1/72° horizontal resolution), as previously presented
in Sauvage et al. (2020). The time step is 60 s.
The set of parameterizations from Ardhuin et al. (2010) is used, as for most of the wave forecasting centres (Ardhuin et al.,
2019). Thus, the swell dissipation is computed with the Ardhuin et al. (2009) scheme, and the wind input parameterization
is adapted from Janssen (1991). Nonlinear wave–wave interactions are computed using the discrete interaction approximation
(Hasselmann et al., 1985). The parameterization of the reflection by shorelines is described in Ardhuin and Roland (2012).
Moreover, the computation of the depth-induced breaking is based on the algorithm from Battjes and Janssen (1978), and the
bottom friction formulation follows Ardhuin et al. (2003).





## 2.2 Air–sea exchanges and coupling

The coupled system AROME-NEMO-WW3 is implemented using the SURFEX-OASIS coupling interface developed by Voldoire et al. (2017). This interface permits the field exchanges between the atmospheric and ocean models on one side and between the atmospheric and wave models on the other side (Fig. 1 and Tab. 1).

NEMO provides to the OASIS3-MCT coupler (OASIS hereafter, Craig et al., 2017) the mean SST and horizontal surface current components ($u_s$ and $v_s$) at the coupling frequency of one hour. At the same coupling frequency, WW3 provides the peak period of the wind sea ($T_p$) to OASIS. These fields, after interpolation onto the AROME (SURFEX) grid, are used to compute surface fluxes at each subsequent atmospheric time step. The wind components of the first atmospheric level ($u_a$,$v_a$) and the air–sea fluxes at the interface - namely the solar heat flux $Q_{sol}$, the non-solar heat flux $Q_{ns}$, the two components of

the horizontal wind stress $\tau_u$ and $\tau_v$ and the atmospheric freshwater flux $EMP$ - are computed by SURFEX and provided to OASIS, which then averages them over one hour, interpolates and sends them to WW3 (for $u_a$ and $v_a$) or NEMO (for $Q_{sol}$, $Q_{net}$, $\tau_u$, $\tau_v$, and $EMP$) at the coupling frequency. Detailed information on the different coupling namelists for each model is given in appendix.

The air–sea fluxes are computed taking into account near-surface atmospheric and oceanic parameters, following the radia-

tive schemes (Fouquart and Bonnel, 1980; Mlawer et al., 1997) and the WASP turbulent fluxes parameterization:

$$Q_{sol} = (1-\alpha)SW_{down} \tag{1}$$

$$Q_{ns} = LW_{down} - \epsilon\sigma\theta_s^4 - H - LE \tag{2}$$

where $SW_{down}$ and $LW_{down}$ are the incoming components of the solar and infrared radiations, respectively. $\theta_s$ is the SST, $\alpha$ is

the albedo, $\epsilon$ is the emissivity and $\sigma$ is the Stefan–Boltzman constant. Turbulent heat fluxes ($H$ for sensible and $LE$ for latent) are calculated with WASP (see the following) and thus depend on the wind speed and on the air–sea gradients of temperature and humidity, respectively, and on transfer coefficients $C_H$ and $C_E$, respectively, that themselves depend on air stability and wave age (see the following).

The atmospheric freshwater flux is given by:

$$EMP = E - P_l - P_s \tag{3}$$

where $E$ is the evaporation, corresponding to $E = LE/\mathcal{L}_v$ with $\mathcal{L}_v$ the vaporization heat constant. $P_l$ and $P_s$ are the liquid and solid surface precipitation rates (given by AROME).

The wind stress takes into account the ocean surface current (given by NEMO), as follows:

$$\boldsymbol{\tau} = (\tau_u, \tau_v) = \rho_a C_D \|\boldsymbol{U_s} - \boldsymbol{U_a}\|(\boldsymbol{U_s} - \boldsymbol{U_a}) \tag{4}$$

with $\rho_a$ the air density, $\boldsymbol{U_a} = (u_a, v_a)$ the wind at the lowest atmospheric model level (around 5 m here) and $\boldsymbol{U_s} = (u_s, v_s)$ the ocean surface current. $C_D$ is the drag coefficient given by the turbulent fluxes parameterization.





The turbulent heat fluxes are also expressed as functions of the air–sea gradients:

$$H = \rho_a c_{pa} C_H \| \boldsymbol{U_s} - \boldsymbol{U_a} \| \Delta\theta$$

$$LE = \rho_a L_v C_E \| \boldsymbol{U_s} - \boldsymbol{U_a} \| \Delta q \tag{5}$$

with $c_{pa}$ the air heat capacity. $\Delta\theta$ and $\Delta q$ represent the air–sea gradients of potential temperature and specific humidity, respectively.

Each transfer coefficient ($C_X$) can be expressed as:

$$C_X = c_x^{\frac{1}{2}} c_d^{\frac{1}{2}} \tag{6}$$

where $X/x$ is $D/d$ for wind stress, $H/h$ for sensible heat and $E/e$ for latent heat. The $c_x^{\frac{1}{2}}$ coefficients are function of $\psi_x(\zeta)$ that describes empirically the stability, $\zeta$ is the $z/L$ ratio with $L$ the Obukhov length, and $z_0$ that is the sea surface roughness length. Therefore:

$$c_x^{1/2}(\zeta) = \frac{c_{xn}^{1/2}}{1 - \dfrac{c_{xn}^{1/2}}{\kappa} \psi_x(\zeta)} \tag{7}$$

and:

$$c_{xn}^{1/2} = \frac{\kappa}{ln(z/z_{0x})} \tag{8}$$

with the subscript $n$ referring to neutral ($\zeta = 0$) stability, $z$ to the reference height and $\kappa$ is Von Karman's constant. The sea surface roughness length $z_0$ is defined by two terms, the Charnock's relation (Charnock, 1955) and a viscous contribution (Beljaars, 1994):

$$z_0 = \frac{\alpha_{ch} u_*^2}{g} + \frac{0.11\nu}{u_*}, \tag{9}$$

with $\nu$ the kinematic viscosity of dry air, the friction velocity $u_*$ and the Charnock coefficient $\alpha_{ch}$. In WASP, $z_0$ depends on the wave age ($\chi$) through the Charnock coefficient ($\alpha_{ch}$) which is a power function of $\chi$, and $\chi$ is defined as:

$$\chi = \frac{gT_p}{2\pi\|\boldsymbol{U_a}\|} \tag{10}$$

where $g$ is the acceleration of gravity and $T_p$ is the peak period of waves corresponding to the wind sea, i.e. the waves generated by the local wind that are growing ($\chi < 0.8$) or in equilibrium with the wind ($0.8 \leq \chi < 1.2$) and that are aligned with the local wind. The reader can refer to Sauvage et al. (2020) for an enlarged description of WASP.

The AROME-France domain is more extended than the NWMED72 domain of NEMO and WW3, and as the Atlantic Ocean and the Adriatic Sea are not represented, there is no air–sea coupling in these areas: the SST comes from the AROME-France initial analysis and is constant during the run, horizontal current is considered null, and the peak period is computed inside WASP as a function of the wind speed ($T_p = 0.5\|\boldsymbol{U_a}\|$).



## 3 Evaluation

### 3.1 Case study

The HPE studied here is described in detail in Sauvage et al. (2020). Its main characteristics are briefly given in the following.

The synoptic situation of the event has been defined as a "cyclonic southerly" kind (Nuissier et al., 2011), characterized by a slow moving trough extending from the British Islands to Spain that induced at upper level a south-westerly flow over South-Eastern France. At low level, a cyclonic circulation established and induced a south-easterly flow across the Western Mediterranean Sea that originated from South-Eastern Tunisia. The event is also marked by a strong easterly flow that orig-

inated from Southern Alps and intensified during the two first phases of the event (Fig. 2). This easterly flow triggered large sea-surface heat exchanges over the Ligurian Sea and along the French Riviera (Fig. 2a,b) due to strong wind (up to 20 m s$^{-1}$ observed at the Azur buoy [7.8°E - 43.4°N]) and to large air–sea gradients. These large fluxes gradually warmed and moistened the low-level air mass along its path towards the Gulf of Lion. The Gulf of Lion was initially affected by the rapid easterly flow, producing a young sea with significant wave height ($H_s$) up to 6 meters and strong air–sea fluxes. As the system moved

eastwards with the highest wind intensity, the sea state evolved in time from a well-developed sea to swell in this region. Throughout the event, the French Riviera was affected by strong easterly wind generating wind sea. The convergence zone between the warm and moist southerly flow and the dry and cold easterly flow was found to trigger convection over the sea. A second convective system, south of France, was initiated by an orographic uplift and was fed by the easterly flow. Both systems produced large amounts of precipitation (Fig. 2c,d).

Four periods of the event were finally distinguished using observations and the atmosphere–wave coupled simulation (hereafter AW, see section 3.2) for the marine low-level conditions and the convective systems life cycle: (I) initiation stage, (II) mature systems, (III) north-eastward propagation and (IV) Tramontane wind onset. In the following, we evaluate the coupling effects during phases I and II.

### 3.2 Numerical set-up

In order to be able to evaluate the contribution of coupling between the different compartments, we set up and compare different numerical experiments. Each experiment is composed of three forecasts of 42 hours range, starting at 00 UTC, on 12, 13, 14 October 2016.

AOW is the ocean–atmosphere–wave coupled simulation using AROME, NEMO and WW3 models. In AOW, no ocean–wave interaction is considered, but the surface fluxes computed with WASP and considered by the three models are perfectly

identical and take into account the interactive evolution of wind, near-surface air temperature and humidity, SST, surface current and wave peak period. The coupling frequency is hourly and the interpolation method is bi-linear (as in the other coupled experiments). The atmospheric initial conditions come from the AROME-France analysis, and in particular the SST field seen by AROME-France outside the northwestern Mediterranean area (NWM hereafter, Fig. 3a). The boundary conditions are provided by the hourly forecast from the Météo-France global model, ARPEGE (Action de Recherche Petite Echelle

Grande Echelle, Courtier et al., 1991). For NEMO-NWMED72, the open boundary conditions come from the global PSY4





daily analyses of Mercator Océan International at 1/12°-resolution (Lellouche et al., 2018). The initial conditions come from a spin-up of NEMO-NWMED72 driven by AROME-France hourly fluxes forecasts (from 0 to +24h each day starting on 5 October 2016) for the forecast starting on 12 October at 00 UTC. For the subsequent forecasts, the ocean initial conditions at 00 UTC (day D) are provided by the AOW (ocean) forecast based on the previous day (D−1; range +24h) through a restart.

The WW3-NWMED72 boundary conditions consist of eight spectral points distributed along the domain and provided by a WW3 global 1/2° resolution simulation (Rascle and Ardhuin, 2013) run at Ifremer. Wave initial conditions are restart files, first from a former WW3 simulation for the forecast starting at 00 UTC on 12 October, then from the previous AOW forecast (D−1; range +24h) for the following days (see Sauvage et al., 2020, for a more detailed description of the wave initial and boundary conditions). Outside the NWM domain, the wave peak period field is estimated as a function of the surface wind and surface

current is considered as null.

An atmosphere–wave coupled simulation (AW) was carried out using AROME and WW3. The initial and boundary conditions for waves and atmosphere are treated as in AOW, the initial SST field comes from the PSY4 daily analysis of the starting day of the forecast and is kept constant throughout the 42 hours of forecast. Surface currents are considered null. Coupling only takes place in the NWM domain. Elsewhere, $T_p$ is computed as a function of the surface wind.

The AO experiment is the coupled ocean–atmosphere simulation, between AROME and NEMO. The initial and boundary conditions for ocean and atmosphere are treated as in AOW. Outside the NWM domain, the SST is given by the AROME-France analyses and the surface current is considered null. Everywhere, $T_p$ is computed as a function of the surface wind.

Two atmosphere-only experiments with AROME-France are also examined, using the same atmospheric boundary and initial conditions as AOW, but different SSTs. In the AY experiment, the SST initial field is taken from the PSY4 daily analyses for

the whole marine domain of AROME-France, whereas in AYSSTatl, the SST forcing comes from the PSY4 analyses only on the NWM domain and from the AROME-France analyses elsewhere. Both AY and AYSSTalt use WASP as turbulent fluxes parameterization with the peak period estimated as a function of the surface wind, a constant SST field during the forecast and null current. Figure 3b shows the differences in SST between the AY and AYSSTatl simulations. The PSY4 SST from an ocean model at 1/12° resolution enables to represent finer structures in the Atlantic Ocean (Fig. 3) compared to the AROME

analysis, which only represents an average structure of the SST field. Differences in the Atlantic Ocean can be as high as 2°C (3°C locally). This simulation is in fact an intermediate simulation justified by the fact that the coupling with NEMO-NWMED72 leads to changes in SST only in the Mediterranean Sea. The comparison between AY and AYSSTatl thus allows for an assessment of the impact of the Atlantic Ocean surface temperature on the HPE forecast.

A summary of the sea-surface conditions for each experiment is given in Table 2. The simulations AY and AW have already

been used and validated in Sauvage et al. (2020), and serve here as references to evaluate the coupling impact.

Note that the insertion of ocean coupling here induce not only a prognostic evolution of the sea surface, but also modifications of the initial SST conditions seen by AROME-France over the NWM domain (Fig. 3c). These differences are induced by both the spin-up strategy and the restart mode of NEMO for each forecast run. Indeed, the spin-up (without assimilation) makes NEMO-NWMED72 slowly diverging from PSY4 but also allows it to produce its own fine-scale structures permitted by its res-

olution (1/72°) and in response to the AROME-France high-resolution atmospheric forcing, whereas directly using the PSY4





3D fields would have let the ocean model adjustment affect the short-range forecast. The choice to restart NEMO for coupled forecasts from the spin-up first, then from a previous forecast was also made to be close to the cycling done in operational context, i.e. using a previous forecast as initial conditions for the surface scheme (and as background for the AROME 3D-Var data assimilation scheme, not done here). This way, the ocean model is initialized with adjusted, fine-scale, and instantaneous

fields, which are representative of ocean conditions in the Mediterranean Sea before the event, while larger-scale daily-mean SST conditions are applied in fact in AY, AYSSTatl and AW with the PSY4 SST analyses.

Thus, regarding the study of Sauvage et al. (2020), the tri-coupling presented here adds new sea surface conditions, with the interactive evolution of the SST and of the currents simulated by NEMO at a kilometric resolution taken into account in

the turbulent fluxes during the HPE forecast. This permits 1) to verify the robustness of the results obtained on wave coupling impact, when an interactive ocean is included, and, 2) to investigate and compare the couplings contributions to HPE forecast.

In order to quantify the impacts of coupling, a sensitivity analysis is conducted by finely analyzing the differences obtained. In particular, the contribution of the tri-coupled system (ocean–atmosphere–waves) will be compared to the impacts of the bi-coupled simulations (i.e. ocean–atmosphere and waves–atmosphere). The method thus consists in comparing the simulations

two by two by estimating the impacts of the coupling (interactive evolution and changes in the initial conditions brought by coupling) on the dynamics (wind) and the low-level environment (temperature, humidity), the turbulent surface fluxes [momentum flux (or wind stress), sensible heat flux $H$ and latent heat flux $LE$], on evaporation and on precipitation. When available, observations of the air–sea interface are also used to qualify the different simulations. The impacts of tri-coupling on the representation of the surface ocean layer and the sea state ($H_s$ and $T_p$) are also examined.


## 4 Coupling impact on forecast

### 4.1 Atmosphere–wave coupling

The analysis of the atmosphere–wave coupling is described in details in Sauvage et al. (2020) with comparison of AW (AWC in Sauvage et al., 2020) to AY. Here are some highlights of the main conclusions.

The main result is a significant increase of the wind stress found along the French Riviera where the low-level wind is the strongest, as taking into account the sea state with the generation of a wind sea leads to an increase in surface roughness. The increase in stress in this region represents +10% during Phase I (between 13 Oct. 03:00 and 18:00 UTC) and +8.6% during Phase II (between 13 Oct. 19:00 UTC and 14 Oct. 03:00 UTC) when compared to AY. The wave coupling has the effect of significantly reducing the wind speed along the French Riviera, up to 3 m.s$^{-1}$ and by 7% in average with notably a decrease

in bias at the Azur buoy. This is reflected in the overall wind speed bias in Table 3 presenting the bias, RMSE (Root Mean Square Error) and correlation coefficient calculated for each experiment with respect to weather surface stations. A spatial shift of about 15 km eastward of convergence line and of heavy precipitation at sea is found, linked to the slow down of the easterly wind upstream (along the French Riviera). In AW, a decrease in latent and sensible heat fluxes was noticed compared





to AY. However, this decrease was only of ∼2% on the total turbulent heat flux, despite a priori favourable conditions for a
larger response (i.e. strong winds, a large air–sea thermal gradient, and a young sea). Wave coupling also leads to significant
differences in the Gulf of Lion, downstream of the convective system over sea, related to internal modifications of the convective
system. Finally, the convective system over the Hérault area appears not sensitive to wave coupling (or forcing). This can be
explained by the fact that orographic uplift is the triggering factor of this system.

Adding the coupling with waves to an AO coupled configuration can impact the heat extraction from the ocean in several
manners (e.g. Renault et al., 2012; Varlas et al., 2020). First, taking into account waves can increase the surface roughness,
leading to larger wind stress and weaker surface wind. This decrease of the wind can directly decrease the heat fluxes (see Eq.
5). Then, the increase of the surface roughness can result in larger transfer coefficients for heat (Eq. 8) that can lead to slightly
larger heat fluxes. Finally, even though the ocean and wave models are not directly coupled in the present study, stronger wind
stress can result in more mixing and cooling in the oceanic surface layer, thus colder SSTs. These colder SSTs can dampen
the turbulent heat fluxes directly and also increase the atmospheric stability at low level, further decreasing the surface wind,
and eventually the turbulent heat fluxes. In the present case, coupling with waves has almost no impact on SST (differences
of less than 0.2°C, not shown). The impact of the $z_0$ increase on the heat transfer coefficients is also negligible (not shown).
Conversely, the decrease of the simulated wind between AO and AOW is comparable to what was obtained between AY and
AW and significant during Phases I and II with differences of more than 1 m s$^{-1}$ over a large area along the French Riviera
(Fig. 4c and Fig. 2a for the location). As a result, latent and sensible heat fluxes are reduced in AOW by 3–4% over Phases I
and II (Figs. 5b and 6c,d) mainly due to the slow down of the wind. This result is in contrast with what was obtained in other
case studies (e.g. Varlas et al., 2020), probably because the surface wind and the mixing in the oceanic mixed layer were much
stronger than here.

Figure 7a presents different probability scores according to 24 hrs precipitation accumulation (between 13 Oct. 00 UTC
and 14 Oct. 00 UTC) thresholds (Ducrocq et al., 2002): ACC (Accuracy), POD (Probability of Detection), FAR (Probability
of False Alarm), FBIAS (Frequency Bias), ETS (Equitable Threat Score) and HSS (Heidke Skill Score) are calculated by
comparison to rain gauge observations shown in Figure 7b. The FAR score is better when it is close to 0, for the others, a score
of 1 is relative to a perfect prediction. Precipitation scores between AOW and AO are close for cumulative thresholds between
0 and 50 mm. More variability appears for higher threshold but overall AO performs better than AOW. The addition of wave
coupling slightly reduces the intensity of precipitation over the Hérault area on average and with a maximum 24 hrs amount in
AOW of 264 mm compared to AO with 306 mm (Tab. 4). Except this punctual decrease in maximum, the heavy rainfall event
over Hérault in AOW is very similar to the one in AO [chronology, area and mean amount, Fig. 8c (see Fig. 2c for location)]
and so there is no degradation due to the inclusion of the wave coupling from a NWP and/or early warning perspective.

For precipitation related to the MCS over sea, the wave coupling induces larger mean values when comparing AOW with
AO (Tab. 4). Figure 8c shows the differences in the 6h accumulation of precipitation at 00 UTC on 14 October between AOW
and AO, i.e. during Phase II. A slight eastward shift of a few km in the location of the precipitation is seen. Since the near-
surface wind in AOW decreases (compared to AO) in the same way as in AW (compared to AY), this shift in the location of the
convergence and heavy precipitation at sea is likely due to the same process, i.e. a higher roughness in the Ligurian Sea and a


slow down of the easterly low-level atmospheric flow.


Comparisons with sea state recorded by moored buoys are additionally used to assess the quality of the wave forecast in AOW and AW simulations. The scores calculated for the sea-state parameters are summarized in Table 5. Few differences in $H_s$ and $T_p$ scores are obtained when comparing AOW to AW, with a reduction in bias for moored buoys, a reduction in RMSE for $T_p$, and a slight decrease of correlation in AOW. The evolution of the sea state during the event is described for three moored

buoys - Tarragona, Lion and Azur - in Figure 9. The $H_s$ time series simulated by AOW and AW are very close. Nevertheless, we observe a trend of increasing values of $H_s$ and $T_p$ in AOW, with for example for $H_s$ +20–40 cm locally in the Gulf of Lion and along the French Riviera that represents an increase of the order of 1–2% on average in these areas.

The differences in $H_s$ are larger around 00 UTC on 14 October, particularly under the convective system. A difference dipole of $\pm 1$ m corresponds in fact to a shift of the maximum $H_s$ values due to the different positioning of the MCS at sea at that

time between AOW and AW. The time series of the wave age during this period show small changes between the simulations (Fig. 9) and we conclude that the characteristics of the sea state forecast remain the same in AW and AOW, with a wind sea (corresponding to wave age < 1) well represented at Lion and Azur.

## 4.2 Atmosphere-Ocean coupling

Figure 3c represents the difference in initial SST in the Mediterranean on 13 October at 00 UTC between AOW and AW (i.e.

the PSY4 analysis). The initial SST is warmer in AOW, especially in the Gulf of Lion, along the French Riviera and in the Tyrrhenian Sea (up to 1.5 °C). On 14 October at 00 UTC, after 24 hours of forecast, the SST in AOW cooled down (Fig. 3d), especially in the Gulf of Lion and along the French Riviera where winds and heat fluxes are the strongest (Fig. 2a,b). In these areas, larger evaporation and latent heat flux are found in AOW compared to AW (+7% in the Azur zone during Phase I, Figs. 5b and 6a,b) due to a warmer SST at the beginning of the event. The sensible heat flux in AOW is also increased by 11%

during phases I and II compared to AW (not shown). This allows more heat and moisture extraction from the ocean mixed layer to the atmospheric low levels and therefore more favourable low-level conditions for convective systems. In the last part of the event, coupling with the ocean results in slightly colder SSTs in AOW than in AW and slightly lower enthalpy fluxes. Ocean coupling appears to have small impact on wind stress and surface wind speed: both simulated parameters in AOW are on average identical to those of AW along the French Riviera and in the Gulf of Lion with differences of less than 0.3 m s$^{-1}$ (Figs.

5a and 4a,d). The largest differences are found in the Gulf of Lion in the form of dipoles that are not homogeneous in time. These patches of differences are mainly due to modifications in the evolution of convective cells and small displacements of the MCS over sea in the different simulations, with consequences on the low-level flow downstream. Same results are observed when comparing heat fluxes and surface dynamics between AO and AYSSTatl. In view of these results, it confirms that ocean coupling including SST and surface currents into the wind stress computation has a very low impact on the near-surface wind

for such strong wind regime largely controlled by the synoptic circulation.

For temperature (T2M) and relative humidity (RH2M) at 2 meters, small differences are obtained on average between the simulations. T2M varies from 1 to 3% on average with a tendency to increase for T2M when the atmosphere is coupled with





the ocean (and/or waves). For RH2M, coupling with the ocean has small impact (< 1%). Although these differences are, on average, not significant, larger differences can be observed at any given time along the French Riviera, and under the convective
system in the Gulf of Lion (not shown).

Coupling with the ocean results in more intense precipitation for the system on the Hérault with a larger mean rainfall amount (Tab. 4 and Fig. 8b) and a maximum 24 hour rainfall amount at 00 UTC on 14 October of 306 mm in AO versus 269 mm in AYSSTatl. This is due to a slightly moister and warmer air mass at low levels over the Gulf of Lion leading to a more intense convection. At sea, an increase in the maximum 24 hour rainfall amount is obtained in AOW (228 mm) compared to
AW (188 mm) [and in AO (196 mm) compared to AYSSTatl (176 mm)], but the mean value remains close. Overall, rainfall scores are better in AO (and AOW) compared to AYSSTatl (and AW) (Fig. 7b). The differences in the 6 hour accumulation of precipitation at 00 UTC on 14 October between AOW and AW appear quite similar to those between AOW and AO, especially for the offshore system (Fig. 8c,d) because of a slight eastward shift of a few km in the location of the precipitation. The effect of ocean coupling on precipitation, although, involves a different mechanism than wave coupling. Indeed, the addition of the
interactive ocean allows for a larger input of heat and moisture due to higher evaporation and heat fluxes during the initiation phase. This leads to an intensification of the system at sea with formation of a cold pool, which reinforces and tends to push eastwards the convergence during the mature phase (Fig. 10).

The strong sensitivity of the convergence at sea to changes in initial SST and to the oceanic feedback was already highlighted by Rainaud et al. (2017) with the AROME-NEMO coupling for another Mediterranean HPE. The present study permits to
identify more clearly the large impact of ocean coupling on heat and water supply, which controls the intensity of convection which itself modifies the MCS motion and location through internal mechanisms acting for this case to a convergence reinforcement.

Concerning ocean forecasts, AOW and AO simulations show very similar results with a positive bias in temperature (0.57°C)
and almost null in salinity (−0.02 psu) when compared to moored and drifting buoy observations between 12 October 00 UTC and 15 October 00 UTC (using the +1h to +24h forecast ranges for each day). The thermohaline characteristics of intermediate and deep waters are very well represented. If we consider only the upper-ocean layer (0–100m), the biases are larger (about −1°C and −0.05 psu, respectively). The most important errors are located between about 15 and 60 meters, with biases up to 6°C and −0.9 psu. These large differences actually reflect an issue in the representation of the thermocline and halocline,
which are deeper but also smoother in the model. Figure 11, comparing the simulated temperature profiles at the Lion and Azur buoys, shows indeed that the mixed layer is thicker and especially that the thermocline is less marked than observed. The same defect of a less marked thermocline [halocline] is found in the analyses of the ocean operational system PSY4 when compared to the same observations, which shows that the biases of AOW and AO are in fact largely inherited from the ocean initial state used. Also, Figure 11 shows the cooling at the Azur buoy under the strong easterly wind observed all along the event (−0.75°C
in 24 hours and −1.4°C in 42 hours observed since 13 October 00 UTC). This ocean response appears quite large considering other HPE studies (e.g. Lebeaupin Brossier et al., 2009, 2014; Rainaud et al., 2017) and is comparable to other high-wind or medicane events (e.g. Renault et al., 2012; Bouin and Lebeaupin Brossier, 2020a). Even though it is significant, it appears to





be underestimated by the model ($-0.6$°C in 24 hours and $-0.85$°C in 42 hours simulated by AOW). Overall, this default in representing the cooling can be explained by the initial ocean state with a too smooth thermocline that limits the mixed-layer

cooling by entrainment, by physical parameters and/or schemes in NEMO and by the absence of ocean–wave coupling.

## 5   Discussion

The comparison of the AOW tri-coupled experiment with the AY atmosphere-only experiment highlights that the combined effect of couplings is an increase in wind stress and enthalpy flux during the initiation and mature phases in the Azur area (Fig. 5 and 6). Here and all along the two phases, the low-level wind is reduced upstream of the offshore MCS (Fig. 4). As

a consequence of larger heat and moisture supplies, both convective systems over Hérault and over sea are more intense and lead to larger precipitation amount forecast (Fig. 8 and Tab. 4). In AOW, the more intense MCS over sea tends to reinforce the convergence (Fig. 10), which is displaced by nearly 100 km eastwards compared to AY.

In fact, the analysis of the coupled simulations AW, AOW, and AO shows the high sensitivity of the location of the heavy precipitating MCS at sea, as an eastward shift of several kilometers of the system is seen with any coupling (Fig. 8). But the

mechanisms identified for this response appear different between wave coupling and ocean coupling. On one hand, the dominant process with wave coupling is the slowing down of the easterly flow due to more roughness that shifts the location of the convergence line, whatever the surface heat flux values are (related to SST or low-level wind variations). On the other hand, ocean coupling (and its initialisation) strongly controls the heat and moisture supply that indirectly impacts the convergence through internal modifications of the convective system [more intense if a higher SST is used during the initiation and mature

stages]. Thus, these results prove the importance and complementarity of both couplings to well represent the complex interactions of the ocean upper- and surface-layer with the marine atmospheric boundary layer, in particular for such severe weather conditions with large exchanges.

The clear splitting between the two couplings impacts on the atmospheric event here has been done thanks to bi-coupled experiments and confirmed in AOW where there is no direct interaction between ocean and waves. However, it has been shown

that surface waves enhanced vertical mixing in the ocean surface layer. In the case of tropical cyclones, Aijaz et al. (2017) for example showed that wave-induced mixing caused significant cooling and a deepening of the mixing layer, which can then impact the intensity of the cyclone. Staneva et al. (2016) and Wu et al. (2019) also showed with sensitivity studies in the North Sea and Baltic Sea that taking into account the effect of waves on the ocean improved surface temperature, ocean surface circulation and sea level height. So, it would be interesting to conduct other experiments by adding the interactive coupling

between ocean and waves, as it would likely modify the turbulence and the exchanges at the air–sea interface. The use of the SURFEX-OASIS coupling interface enables to quickly consider the insertion of the full coupling between NEMO and WW3, as recently developed by Couvelard et al. (2020) with updates in the physics of NEMO (v3.6) and validated through a global coupled modelling study. As mentioned in Section 4.2, the SST initial field is of great importance for short term forecast of extreme events involving large air–sea fluxes (e.g. Lebeaupin Brossier et al., 2009; Rainaud et al., 2017)). The spin-up strategy

used to start NEMO in the AO and AOW coupled simulations induced large discrepancies when compared to the PSY4 daily



analysis (Fig.3) as in AROME-only simulations (AY or AYSSTatl). On the other hand, the use of the PSY4 SST daily analysis to start the forecast means that initial conditions (i.e. at 00 UTC) are actually a 24 h average of the SST including changes of SST due to the studied event. To better illustrate this initialisation issue, the comparison of the 6 m-depth temperature at Azur in Figure 11 shows that starting with the PSY4 analysis on 13 October leads to a significant initial cold bias compared
to observations ($-0.5°C$, similarly for SST) as PSY4 already accounts for the cooling during that day. For atmosphere-only or atmosphere–waves forecasts, the SST bias reduces with time, while this error would have persisted then if it has been used to initiate NEMO in the AO and AOW coupled experiments. So, for coupled forecast of HPEs or other severe weather events happening over a short period of time (< 24 hours), an ocean initial state corresponding to a hourly average for example or to an instantaneous state is preferable to avoid this potential bias prolongation.

Moreover, we investigated the influence of the Atlantic Ocean surface conditions on the AROME forecast by comparing AY and AYSSTatl. As expected, the Atlantic Ocean SST differences between AY and AYSSTatl have a small impact on low-level conditions in the Mediterranean area, as the latent heat flux and the wind stress are on average identical, especially along the French Riviera (Fig. 5). The scores in Table 3 confirm that AY and AYSSTatl are similar for precipitation forecast. The scores of AY and AYSSTatl are also close for thresholds between 0 and 50 mm. We note more variations for larger rainfall amounts,
with overall a slight improvement in AY (when SST from PSY4 is used in Atlantic rather than the AROME analysis). Thus, the difference in SST over the Atlantic Ocean has a very small impact. Indeed, for this event driven mainly by eastern and southern flows that supply MCS in heat and moisture extracted from the Mediterranean Sea, the change in SST in the Atlantic has a small influence on these low-level flows. However, the change in SST may have had an impact on the position of the cold front and disturbed the convergence affecting, in particular, the formation and movement of the MCS at sea, which may
explain the slightly larger differences found in the Gulf of Lion.

Regarding the ocean surface current coupling, recent studies highlighted the importance of the representation of the current–wind interactions and the atmosphere feedback on the ocean mesoscale structures (e.g. Seo et al., 2016; Seo, 2017; Renault et al., 2016; Jullien et al., 2020). Renault et al. (2017, 2019) showed a damping of the eddy kinetic energy due to the current feedback modulation of the energy transfer between the ocean and the atmosphere leading to more realistic simulations. These
current–wind interactions need to be further investigated in our coupled system with the insertion of the current terms in the AROME turbulence scheme. However, in this particular HPE case, as the near-surface wind speed is largely superior (> 20 m s$^{-1}$) to the surface current velocity (< 1 m s$^{-1}$), we hypothesize that the feedback of the surface current on the atmosphere might be small (as in Bouin and Lebeaupin Brossier (2020b) for instance).

The numerical performances of the various simulations are finally briefly presented in Table 6 for 42 h range forecasts. First, it is important to note that only AY uses the AROME uncoupled binary and the AROME input/output (I/O) server, with a distribution of 8 processes by core (i.e. 48 cores of 8 processes for AROME and 2 cores of 8 processes for its I/O server). The AYSSTatl simulation is an atmosphere-only simulation but is related to a toy model through the SURFEX-OASIS interface in order to initiate SST from two various sources (AROME and PSY4 analyses) as imitating an ocean model. Also, the AROME
I/O server is switched off in AYSSTatl (as for all simulations using OASIS) because the MPI (Message Passing Interface) link





between OASIS and the AROME I/O server is not inserted yet. The choice was made to always keep the distribution of 8 processes by core, and thus the toy model allocates one core of 8 processes, while AROME keeps 48 cores of 8 processes (i.e. 384 processes in total). The comparison of AYSSTatl with AY shows an increase in the time elapsed (+22%), and in the total central processing unit (CPU) cost (+8.5%) and a large loss of efficiency (shown by the CPU time values) due to the fact that

the toy model processes are "sluggish" all the forecast long and also due to the cost of undistributed I/O task.

Hereafter, the coupled forecasts are compared to AYSSTatl. The ocean coupling in AO increases the total CPU cost by 1.6% with only 2 cores of 8 processes allocated for NEMO, and the time elapsed by 22%. This latter increase is in fact related to the rebuild task that reassociates the NEMO output files of each process in a single file containing the whole NWMED72 domain. For future versions of the coupled system, this will be completely avoided with the use of the XIOS library (*XML-IO-Server*,

Meurdesoif, 2013) to manage the NEMO outputs. Except this, the ocean coupling can be considered as very light in terms of computing cost. The wave coupling in AW is done with 6 cores of 8 processes for WW3 and one core to manage the SST field with a toy model. It increases the total CPU time by 13.8% and the time elapsed by 57% in respect to AYSSTatl. Considering the delivery constraint in an operational forecasting system this represents a significant increase in time. However, it must be said here that no specific effort have been made to balance the various computation times, although possible using a higher number

of processes for WW3, and to optimize the calculation time on the Météo-France High Performance Computing system (HPC) and thus, improvement in this matter needs to be done, in particular concerning WW3 compilation options. Finally, AOW shows both increases in elapsed time (and Integrated Elapsed Time (IET)] and in total CPU cost consistent with the addition of the two couplings.

## 6   Conclusions

This study presents the ocean–atmosphere–wave coupled system, developed using the NWP model AROME, the NEMO ocean circulation model and the wave model WW3, all at a kilometric resolution. This system is designed to better understand and represent the exchanges at the air–sea interface and to evaluate the impact on the weather forecast using a case study corresponding to a Mediterranean HPE that occurred in mid-October 2016. In order to quantify the contributions of the different couplings, a set of bi-coupled and tri-coupled simulations were carried out. Sensitivity analysis highlighted the importance of

coupling with waves on the dynamics of the lower levels of the atmosphere. Indeed, the slow down of the near surface wind along the French Riviera occurring in AW is preserved in the same proportions in the AOW tri-coupled experiment. Compared to these results, the coupling with an interactive ocean appears to have small impact on the momentum flux and on the surface wind. Nevertheless, the coupling with the ocean plays an important role on air–sea heat exchanges. The warmer ocean in AO increases heat and moisture extraction and therefore changes the development of the convective systems. This also affects the

convergence line at sea with the establishment of a better organized system. Regarding the heavy precipitation over the Hérault region, we observe a weak variability through the different simulations which can be explained by its triggering mechanism that is mainly controlled by orographic uplifting. The offshore system shows a greater sensitivity to coupling with, in particular, displacements of the convergence line inducing differences in intensity and location of the heavy precipitation.





The validation of the ocean compartment with in-situ observations showed a good representation of the near-surface ocean
layer and showed no significant impact due to wave coupling in AOW. The validation of the wave compartment, when com-
paring AW and AOW, also showed little differences despite a decrease of the bias (and RMSE) for $T_p$ in the AOW simulation.
These results permit to be confident in the numerical and scientific benefits of coupling ocean and wave forecasts to atmosphere
even for short-range forecast and in the feasibility of integrated forecasts.

More generally, the current development of high-resolution coupled models allows to resolve phenomena at a kilometric
scale. The recent deployments of new airborne or spaceborne observing capabilities enable to detect very fine structures at
the sea surface (sharp SST fronts, filaments, strong contrasts of currents for instance) thanks to their signature on the surface
roughness (e.g. Rascle et al., 2017; Wang et al., 2019). These surface sub-kilometric features of oceanic or meteorological
origin are likely present as small-scale modulations of larger-scale gradient of SST, surface current or wave field. Oceanic
modelling is now able to accurately represent such structures and their time evolution, provided the resolution of the simulation
is fine enough (e.g. Gula et al., 2014). SST fronts for instance can significantly impact the atmospheric conditions (Small
et al., 2008), low-level flow (Redelsperger et al., 2019) and convergence (Meroni et al., 2020) independently of strong-impact
weather events. The feasibility of using tri-coupled configurations like the one developed in the present study for a reasonable
computing cost opens the way to a more explicit representation of the surface heterogeneities at sea, of their time evolution,
and of their impact on the atmosphere for high-resolution deterministic operational NWP. If coupling allows more realism,
the quality of coupled forecasts remains however still constrained by the resolution of computations, by the approximations in
some physical processes parametrizations and by the shortcomings of the observing systems initializing the different numerical
models involved. And so, to carefully separate a predictive value from the noise related to coupled forecast errors, further
studies need also to be conducted to examine the propagation of uncertainties in a coupled system through ensemble coupled
experiments, which are now within our reach, for a larger number of cases covering a larger range of weather situations.

*Code and data availability.* Although the operational AROME code cannot be obtained, the modified sources for cy41 are available on
demand to the authors for the partners of the ACCORD consortium and will be included in the cycle 48 Météo-France official release. The
source codes of the other components are available online:

 – WaveWatchIII was used in version 5.16 which is distributed under an open-source style license through a password-protected distribu-
tion site at https://polar.ncep.noaa.gov/waves/wavewatch/. Since version 6.07, WaveWatchIII is distributed using GitHub
(https://github.com/NOAA-EMC/WW3) without any username and password required to access the software package;

 – NEMO is available at https://www.nemo-ocean.eu/ after a user registration on the NEMO website. The version used is NEMO_v3.6_STABLE
for Mediterranean configurations (see https://sourcesup.renater.fr/wiki/morcemed/nemconfig and Appendix);

 – OASIS3-MCT was used in version OASIS3-MCT_3.0. It can be downloaded at https://portal.enes.org/oasis. The public may copy,
distribute, use, prepare derivative works and publicly display OASIS3-MCT under the terms of the Lesser GNU General Public License
(LGPL) as published by the Free Software Foundation, provided that this notice and any statement of authorship are reproduced on all
copies;



- and SURFEX open-source version (Open-SURFEX) including the interface with OASIS from v8_0 is available at http://www.umr-cnrm.fr/surfex/ using a CECILL-C Licence (a French equivalent of the L-GPL licence; http://www.cecill.info/licences/Licence_CeCILL-C_V1-en.txt), but with exception of the gaussian grid projection, the LFI and FA I/O formats, and the dr HOOK tool. The sources for wave–atmosphere coupling within the SURFEX-OASIS interface and the WASP parameterization will be included in the next release (v9) of SURFEX, but can be provided on demand by the authors for older SURFEX versions (back to v7_3).

Outputs from all simulations discussed here are available upon request to the authors.

The Antilope product can be available for research purposes upon request (contact: olivier.laurantin@meteo.fr). The surface weather station data and the chains of thermistors on the Lion and Azur Météo-France's moored buoys are available on the MISTRALS/HyMeX database (mistrals.sedoo.fr) after subscription. Oceanographic buoys data and the PSY4V3R1 daily analyses of Mercator Ocean International are available trough the Copernicus Marine Environment Monitoring Service (CMEMS) portal (marine.copernicus.eu) after user registration.

*Author contributions.* All authors (CS, CLB, and MNB) contributed to the conceptualization and methodology of the study as well as drafting, reviewing and editing the article. MNB developed the WASP parameterization and managed its integration into the SURFEX code. The configuration NWMED72 of both NEMO and WaveWatchIII models were developed and coupled to AROME by CS and CLB. Simulations were ran by CS. CS, CLB and MNB carried out the validation and analysis of the results.

*Competing interests.* The authors declare that they have no conflict of interest.

*Acknowledgements.* This work is a contribution to the HyMeX program (Hydrological cycle in the Mediterranean EXperiment - www.hymex.org) through INSU-MISTRALS support. The authors acknowledge the Occitanie French region for its contribution to César Sauvage's PhD at CNRM. The authors gratefully acknowledge Véronique Ducrocq [Department of Operations for Prediction (DirOP) of Météo-France & CNRM] for motivating, carefully following and promoting this work and for her encouragements during the redaction process. The authors acknowledge the MISTRALS/HyMeX database teams (ESPRI/IPSL and SEDOO/OMP) for their help in accessing to the surface weather station data and the chains of thermistors on the Lion and Azur Météo-France's moored buoys. The authors finally thank Olivier Laurantin from the Observing System Department (DSO) of Météo-France who provided the Antilope product.

## Appendix A: Namelists summary for coupling

Tables A1, A2, A3 and A4 specify the parts inserted in the various namelists for the AOW coupled simulation. The reader can also refer to the users documentations of SURFEX, WaveWatchIII, NEMO and OASIS.



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





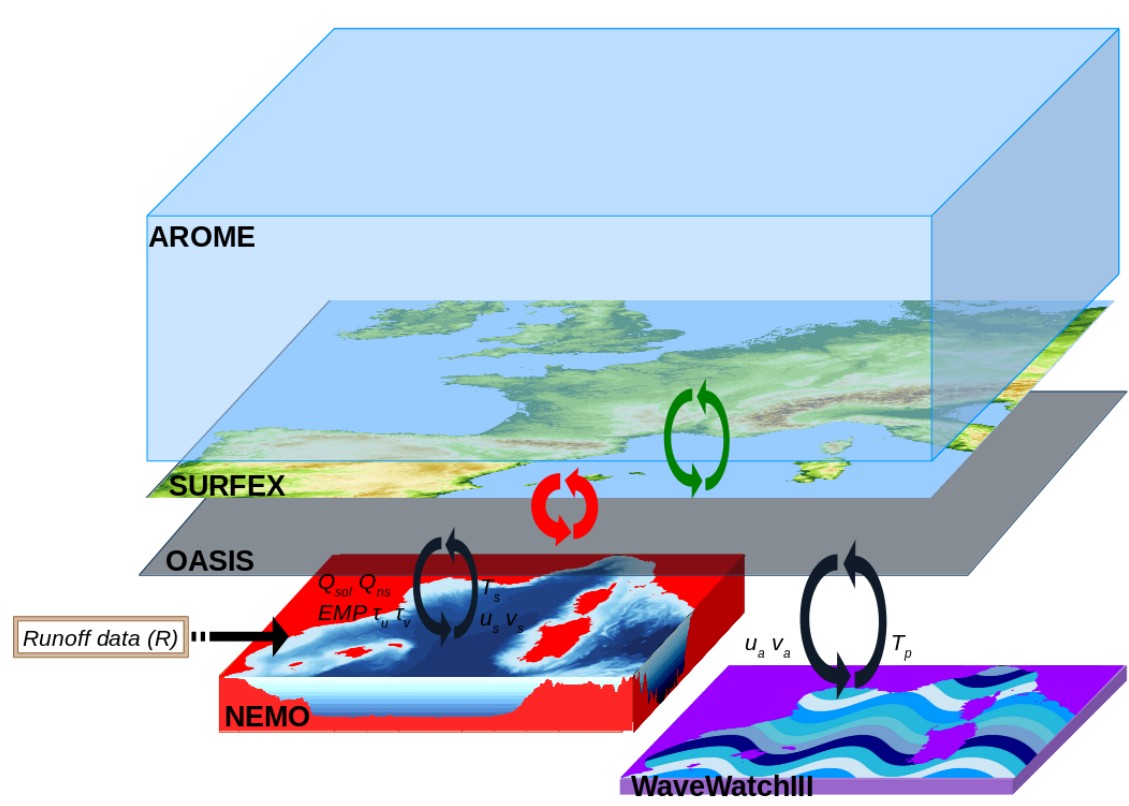

**Figure 1.** The NEMO-AROME-WW3 coupled architecture and domains illustrated by orography (of the AROME-France domain in the SURFEX "area") and the NWMED72 bathymetry (in the NEMO box). The SURFEX-OASIS interface (red arrows) is detailed in Voldoire et al. (2017) and the AROME-SURFEX links (green arrows) are described in Masson et al. (2013) and Seity et al. (2011). See text and Table 1 for the exchanges involving NEMO and WW3.



**Figure 2.** Mean surface and atmospheric low-level conditions: (a,b) enthalpy flux over sea ($H + LE$, colors, W m$^{-2}$), Convective Available Potential Energy (CAPE, green contours every 750 J kg$^{-1}$) and 10 m-wind (arrows, m s$^{-1}$) and (c,d) $\theta_w$' (colors, K) and wind (arrows, m s$^{-1}$) at 925 hPa, and total rainfall amounts (green contours every 50 mm) from the AW forecast during (a,c) the initiation phase (Phase I, between 13 Oct. 2016 03 UTC and 18 UTC) and (b,d) the mature phase (Phase II, between 13 Oct. 2016 19 UTC and 14 Oct. 2016 03 UTC). See text and Sauvage et al. (2020) for more details. The dashed purple box in (a) indicates the Azur zone. The dashed boxes in (c) indicates the Hérault (purple) and offshore (cyan) areas for precipitation analyse.

(a)

(b)

(c)

(d)

**Figure 3.** (a) SST (°C) forecast in AOW for 13 October at 14 UTC (forecast basis: 13 Oct. 00 UTC) and (b) differences in initial SST fields (°C, 13 Oct. 00 UTC) between AY and AYSSTatl (AROME forecasts with persistent SST, see text and Tab. 2). Comparison of the AOW SST forecast (basis: 13 Oct. 00 UTC) over NWM for (c) 13 October 01 UTC and (d) 14 October 00 UTC, with the PSY4 daily analysis of 13 October (used in AY/AYSSTatl/AW experiments).



(a)

(b) AY-AYSSTalt

(c) AOW-AO

(d) AOW-AW

**Figure 4.** (a) Time series of the surface wind (m s$^{-1}$) forecasts on average over the Azur area, and differences in surface wind for 14 October 00 UTC between (b) AY and AYSSTatl, (c) AOW and AO and (d) AOW and AW (forecast basis: 13 Oct. 00 UTC).

(a)

(b)

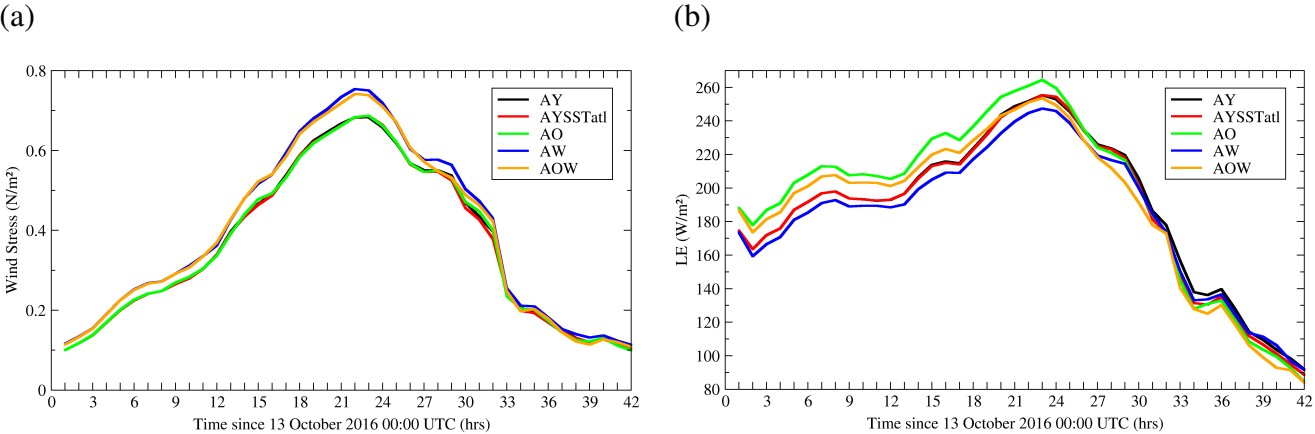

**Figure 5.** Time series of (a) wind stress (N m$^{-2}$) and (b) latent heat flux ($LE$, W m$^{-2}$) on average over the Azur area, for forecasts starting on 13 October 2016 00 UTC.

(a) AOW-AW    13 OCT 14 UTC

(b) AOW-AW    14 OCT 00 UTC

(c) AOW-AO    13 OCT 14 UTC

(d) AOW-AO    14 OCT 00 UTC

**Figure 6.** $LE$ differences (W m$^{-2}$) (a,c) on 13 October at 14 UTC and (b,d) on 14 October at 00 UTC between AOW and AW experiments (a,b) and between AOW and AO experiments (c,d).

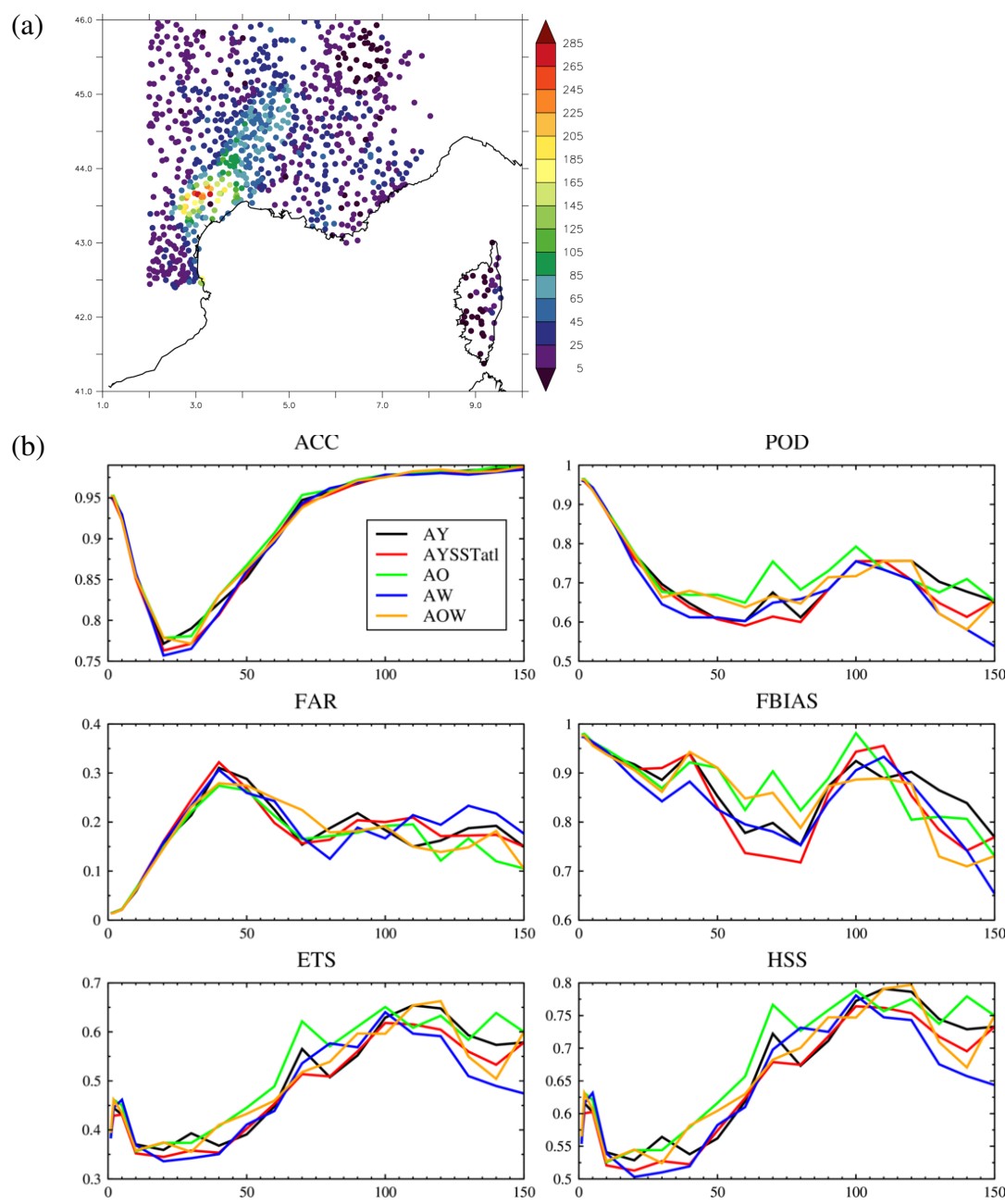

**Figure 7.** (a) Locations and measurements of 24 hours cumulative precipitation (mm) on 14 October at 00 UTC of the Météo-France rain gauges over the south-eastern quarter of France. (b) Forecasts skill scores against rain-gauges observations calculated for cumulative rainfall in 24 hours on 14 October 00 UTC. The $x$-axis indicates the rainfall threshold considered, in mm.

(a) AY-AYSSTatl

(b) AO-AYSSTatl

(c) AOW-AO

(d) AOW-AW

**Figure 8.** Differences in 6 hours cumulative precipitation (mm) on 14 October at 00 UTC (a) between AOW and AO and (b) between AOW and AW.



**Figure 9.** Time series of simulated significant wave height $H_s$ and wave age at the three moored buoys Tarragona, Lion and Azur, from 12 October 00 UTC to 15 October 00 UTC, using successive forecasts of each experiment including WW3 (+1 – +24h forecast ranges each day).





**Figure 10.** (a,c,e) Wind divergence ($10^{-3}$s$^{-1}$) at 950 hPa, vertical velocity (Pa s$^{-1}$, black contours) at 950 hPa and surface wind at 925 hPa (m s$^{-1}$, arrows). (b,d,f) $\theta'_w$ at 925 hPa (°C), CAPE (> 750 J kg$^{-1}$, dark blue line), and surface wind at 925 hPa (m s$^{-1}$, arrows) and reflectivities at 2000 m (dBz, light blue line) on 14 October 14 at 00 UTC for (a,b) AYSSTatl, (c,d) AO and (e,f) AOW.



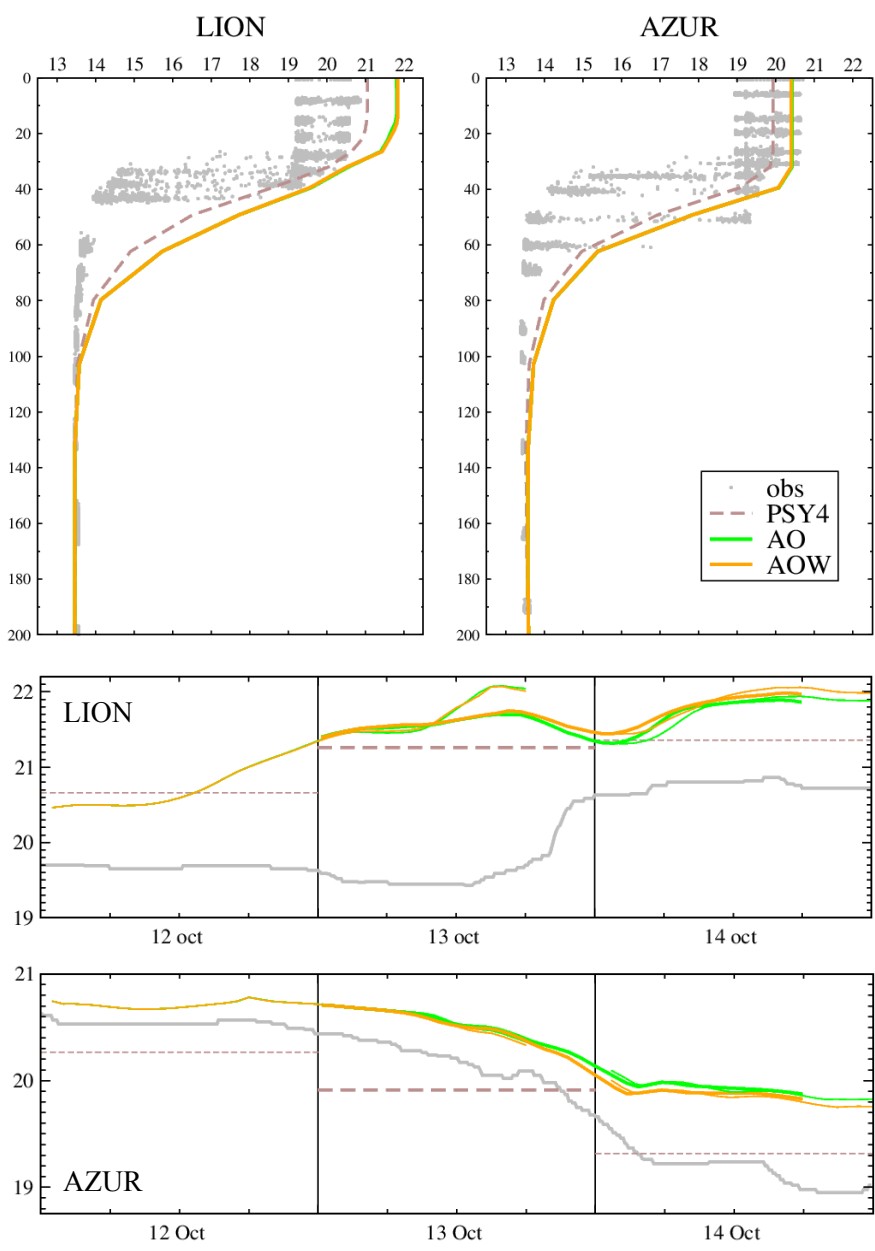

**Figure 11.** Upper panels: ocean temperature profiles (°C) observed by the chains of thermistors at the Lion (left) and Azur (right) buoys between the 12 and the 14 October (grey dots) and simulated by AO and AOW on average for the day of 13 October. Temperature profiles in the PSY4 operational system analysis of 13 October are shown in dashed lines. Lower panels: 6 m depth ocean temperature time series (°C) observed at Lion and Azur and simulated by AO and AOW (successive forecasts). The dashed lines correspond to the values in PSY4.


**Table 1.** List of the exchanged fields.

| *SOURCE model to TARGET model* | |
| --- | --- |
| *Annotation* | *Field description* |
| **NEMO to AROME/SURFEX** | |
| $\theta_s$ | Sea Surface Temperature |
| $u_s$ | Sea surface zonal current |
| $v_s$ | Sea surface meridian current |
| **AROME/SURFEX to NEMO** | |
| $\tau_u$ | Zonal component of the wind stress |
| $\tau_v$ | Meridian component of the wind stress |
| $Q_{ns}$ | Non solar heat flux |
| $Q_{sol}$ | Solar net heat flux |
| $EMP$ | Freshwater flux |
| **WW3 to AROME/SURFEX** | |
| $T_p$ | Wind-sea peak period |
| $H_s$ | Significant wave height (not used in WASP) |
| **AROME/SURFEX to WW3** | |
| $u_a$ | zonal wind at first level |
| $v_a$ | meridian wind at first level |





**Table 2.** Summary of the simulations. Outside the North-Western Mediterranean (NWM) area, surface current is always null and $T_p$ is a function of the wind ($U_a$) only.

| | models | SST (outside NWM) | SST (over NWM) | currents (over NWM) | $T_p$ (over NWM) |
|---|---|---|---|---|---|
| **AY** | AROME | PSY4 | | null | $f(U_a)$ |
| **AYSSTatl** | AROME | AROME analysis | PSY4 | null | $f(U_a)$ |
| **AO** | AROME-NEMO | AROME analysis | **coupled**<br>initially: NEMO spin-up for 12 Oct.,<br>then AO D−1 +24h forecast | | $f(U_a)$ |
| **AW** | AROME-WW3 | PSY4 | | null | **coupled** |
| **AOW** | AROME-NEMO-WW3 | AROME analysis | **coupled**<br>initially: NEMO spin-up for 12 Oct.,<br>then AOW D−1 +24h forecast | | **coupled** |





**Table 3.** Scores against observations from moored buoys and surface weather stations for the 10 m wind speed (WSP, m s−1), the 10 m wind direction (WDIR, °), the air temperature at 2 m (T2M, °C) and the relative humidity at 2 m (RH2M, %).

|  | WSP | | | WDIR | | | T2M | | | RH2M | | |
|---|---|---|---|---|---|---|---|---|---|---|---|---|
|  | Bias | RMSE | Corr. | Bias | RMSE | Corr. | Bias | RMSE | Corr. | Bias | RMSE | Corr. |
| **AY** | 0.22 | 2.70 | 0.66 | 1.43 | 42.05 | 0.85 | 0.39 | 1.25 | 0.70 | 2.19 | 8.84 | 0.79 |
| **AYSSTatl** | 0.24 | 2.69 | 0.66 | 1.29 | 42.61 | 0.86 | 0.4 | 1.25 | 0.71 | 2.24 | 8.88 | 0.78 |
| **AO** | 0.28 | 2.74 | 0.65 | 2.65 | 42.14 | 0.85 | 0.53 | 1.34 | 0.71 | 1.97 | 9.03 | 0.77 |
| **AW** | 0.09 | 2.67 | 0.65 | 1.85 | 42.95 | 0.85 | 0.44 | 1.32 | 0.66 | 3.0 | 9.97 | 0.76 |
| **AOW** | 0.1 | 2.71 | 0.65 | 1.99 | 42.8 | 0.88 | 0.57 | 1.4 | 0.67 | 2.55 | 9.8 | 0.75 |

**Table 4.** Simulated maximum and mean values of rainfall amounts (mm) in 24 hrs, on the October 14 at 00 UTC, over the Hérault zone and the offshore zone around MCSs for the different experiments (forecast starting on the October 13 at 00 UTC).

|  | Zone 1 (Hérault) | | Zone 2 (Sea) | |
|---|---|---|---|---|
|  | Maximum | Mean | Maximum | Mean |
| **AY** | 273.4 | 58.8 | 214.1 | 42.2 |
| **AYSSTatl** | 269.7 | 57.2 | 176.5 | 42.4 |
| **AO** | 306.2 | 60.9 | 196.5 | 43.5 |
| **AW** | 271.9 | 56.8 | 188.1 | 43.5 |
| **AOW** | 264.6 | 58.4 | 228.8 | 45.1 |
| **ANTILOPE** | 287.9 | 73.2 | *348.2* | *51.6* |





**Table 5.** Scores against wave observations from moored buoys and satellites for $H_s$ (m) and $T_p$ (s).

|  | Moored buoys | | | | | | Satellites | | |
|---|---|---|---|---|---|---|---|---|---|
|  | $H_s$ | | | $T_p$ | | | $H_s$ | | |
|  | Bias | RMSE | Corr. | Bias | RMSE | Corr. | Bias | RMSE | Corr. |
| **AW** | −0.28 | 0.58 | 0.90 | −1.27 | 1.64 | 0.88 | −0.28 | 0.5 | 0.71 |
| **AOW** | −0.22 | 0.61 | 0.89 | −0.87 | 1.34 | 0.85 | −0.28 | 0.5 | 0.72 |

**Table 6.** Computation scaling on Météo-France HPC for a 42h-range forecast.

| Experiment | nb procs | | | time | IET | CPU time | Total CPU |
|---|---|---|---|---|---|---|---|
|  | AROME | NEMO | WW3 | elapsed |  |  | cost |
|  | 1440×1536×90 | 933×657×50 | 933×657 |  |  |  |  |
|  |  | (or *toymodel*) |  |  |  |  |  |
| AY | 384 (+16 for ioserv) | - | - | 2:10:29 | 181-05:26:40 | 36-03:03:00 | 199-13:39 |
| AYSSTatl | 384 | *8* | - | 2:39:13 | 219-07:08:40 | 5-10:01:37 | 216-17:28 |
| AO | 384 | 16 | - | 3:15:31 | 271-13:13:20 | 5-15:59:00 | 220-02:51 |
| AW | 384 | *8* | 48 | 4:08:47 | 380-02:03:20 | 9-08:03:50 | 246-15:39 |
| AOW | 384 | 16 | 48 | 4:30:12 | 420-07:28:00 | 9-10:23:00 | 247-08:45 |





**Table A1.** SURFEX namelist (EXSEG1.nam) parameters used for coupling (AOW experiment).

| $NAM_OASIS | |
| --- | --- |
| LOASIS | .TRUE. |
| CMODEL_NAME | 'aromex' |

| $NAM_SEAFLUXN | |
| --- | --- |
| CSEA_FLUX | 'WASPV1' |
| LPWG | .TRUE. |
| LPRECIP | .TRUE. |
| LPWEBB | .TRUE. |
| CSEA_ALB | 'TA96' |
| XICHCE | 0. |

| $NAM_SFX_SEA_CPL | |
| --- | --- |
| XTSTEP_CPL_SEA | 3600. |
| CSEA_FWSU | 'ASFXTAUX' |
| CSEA_FWSV | 'ASFXTAUY' |
| CSEA_HEAT | 'ASFX_QNS' |
| CSEA_SNET | 'ASFX_QSR' |
| CSEA_WIND | ' ' |
| CSEA_FWSM | ' ' |
| CSEA_EVAP | ' ' |
| CSEA_RAIN | ' ' |
| CSEA_SNOW | ' ' |
| CSEA_WATF | 'ASFX_WAT' |
| CSEA_SST | 'ASFX_SST' |
| CSEA_UCU | 'ASFXUCUR' |
| CSEA_VCU | 'ASFXVCUR' |

| $NAM_SFX_WAVE_CPL | |
| --- | --- |
| XTSTEP_CPL_WAVE | 3600. |
| CWAVE_U10 | 'ASFX_U10' |
| CWAVE_V10 | 'ASFX_V10' |
| CWAVE_CHA | ' ' |
| CWAVE_UCU | ' ' |
| CWAVE_VCU | ' ' |
| CWAVE_TP | 'ASFX__TP' |
| CWAVE_HS | 'ASFX__HS' |

| $NAM_DIAG_SURFN | |
| --- | --- |
| LSURF_BUDGET | .TRUE. |
| N2M | 2 |
| LRAD_BUDGET | .TRUE. |
| LCOEF | .TRUE. |





**Table A2.** Part dedicated to coupling in the WaveWatch3 namelist (ww3_shel.inp) for the forecast starting on 13 oct. 2016 00 UTC (AOW experiment).

```
$ Type 7 : Coupling (must be fully commented if not used)
$        Diagnostic fields to exchange (same format as output fields)
$
  20161013 000000   3600   20161014 180000
  N
$
$   - Sent fields by ww3:
$      - Ocean model : T0M1 HS DIR BHD TWO UBR FOC TAW LM DRY
$      - Atmospheric model : CUR CHA HS FP
$
  FWS AHS
$
$   - Received fields by ww3:
$      - Ocean model : SSH CUR DRY
$      - Atmospheric model : WND
$
  WND
$
```


**Table A3.** NEMO namelist (namelist_cfg) parameters used for coupling (AOW experiment).

| $namsbc | |
|---|---|
| nn_fsbc | 5 |
| ln_ana | .false. |
| ln_flx | .false. |
| ln_blk_clio | .false. |
| ln_blk_core | .false. |
| ln_blk_mfs | .false. |
| ln_cpl | .true. |
| ln_mixcpl | .false. |
| nn_components | 0 |
| ln_apr_dyn | .false. |
| nn_ice | 0 |
| nn_ice_embd | 1 |
| ln_dm2dc | .false. |
| ln_rnf | .true. |
| nn_isf | 0 |
| ln_ssr | .false. |
| nn_fwb | 0 |
| ln_wave | .false. |
| ln_cdgw | .false. |
| nn_lsm | 0 |
| nn_limflx | -1 |

| $namsbc_cpl | |
|---|---|
| sn_snd_temp | 'oce only' , 'no' , '' , '' , '' |
| sn_snd_alb | 'none' , 'no' , '' , '' , '' |
| sn_snd_thick | 'none' , 'no' , '' , '' , '' |
| sn_snd_crt | 'oce only' , 'no' , 'spherical' , 'eastward-northward' , 'T' |
| sn_snd_co2 | 'none' , 'no' , '' , '' , '' |
| sn_rcv_w10m | 'none' , 'no' , '' , '' , '' |
| sn_rcv_taumod | 'none' , 'no' , '' , '' , '' |
| sn_rcv_tau | 'oce only' , 'no' , 'spherical' , 'eastward-northward', 'T' |
| sn_rcv_dqnsdt | 'none' , 'no' , '' , '' , '' |
| sn_rcv_qsr | 'oce only' , 'no' , '' , '' , '' |
| sn_rcv_qns | 'oce only' , 'no' , '' , '' , '' |
| sn_rcv_emp | 'oce only' , 'no' , '' , '' , '' |
| sn_rcv_rnf | 'climato' , 'no' , '' , '' , '' |
| sn_rcv_riv | 'none' , 'no' , '' , '' , '' |
| sn_rcv_cal | 'none' , 'no' , '' , '' , '' |
| sn_rcv_co2 | 'none' , 'no' , '' , '' , '' |
| sn_rcv_iceflx | 'none' , 'no' , '' , '' , '' |
| nn_cplmodel | 1 |
| ln_usecplmask | .false. |





**Table A4.** OASIS namelist (namcouple) details for the AOW experiment: *torc* or *tww3* is the NWMED72 grid name, *taro* is the full AROME-France grid name, and *tame* is the AROME-France grid name masked (to land) outside the north-western Mediterranean sea domain.

| source field name (grid/mask) | target field name (grid/mask) | LAG | LOCTRANS | MAPPING | coupling frequency |
|---|---|---|---|---|---|
| O_SSTSST (torc) | ASFX_SST (tame) | 0 | INSTANT | nwmed72_to_aromefr-med_BILINEAR | 3600. |
| O_OCurx1 (torc) | ASFXUCUR (tame) | 0 | INSTANT | nwmed72_to_aromefr-med_BILINEAR | 3600. |
| O_OCury1 (torc) | ASFXVCUR (tame) | 0 | INSTANT | nwmed72_to_aromefr-med_BILINEAR | 3600. |
| ASFXTAUX (taro) | O_OTaux1 (torc) | 50 | AVERAGE | aromefr_to_nwmed72_BILINEAR | 3600. |
| ASFXTAUY (taro) | O_OTauy1 (torc) | 50 | AVERAGE | aromefr_to_nwmed72_BILINEAR | 3600. |
| ASFX_QNS (taro) | O_QnsOce (torc) | 50 | AVERAGE | aromefr_to_nwmed72_BILINEAR | 3600. |
| ASFX_QSR (taro) | O_QsrOce (torc) | 50 | AVERAGE | aromefr_to_nwmed72_BILINEAR | 3600. |
| ASFX_WAT (taro) | OOEvaMPr (torc) | 50 | AVERAGE | aromefr_to_nwmed72_BILINEAR | 3600. |
| WW3__FWS (tww3) | ASFX__TP (tame) | 60 | AVERAGE | nwmed72_to_aromefr-med_BILINEAR | 3600. |
| WW3__AHS (tww3) | ASFX__HS (tame) | 60 | AVERAGE | nwmed72_to_aromefr-med_BILINEAR | 3600. |
| ASFX_U10 (taro) | WW3__U10 (tww3) | 50 | INSTANT | aromefr_to_nwmed72_BILINEAR | 3600. |
| ASFX_V10 (taro) | WW3__V10 (tww3) | 50 | INSTANT | aromefr_to_nwmed72_BILINEAR | 3600. |

$NFIELDS is set to 12, $RUNTIME to 151200 and the line for $NBMODEL is '3 aromex oceanx wwatch 99 99'