# Peer review of "Towards kilometer-scale ocean—atmosphere—wave coupled forecast: a case study on a Mediterranean heavy precipitation event"

_Atmospheric Chemistry and Physics, 2021_

## Author Comment (AC1)

Reply to Reviewers

manuscript number : **acp-2021-239**
entitled : "*Towards kilometer-scale ocean–atmosphere–wave coupled forecast : a case study on a Mediterranean heavy precipitation event*"
authors : Sauvage, C., Lebeaupin Brossier, C., Bouin, M.-N.
* * *
**Reviewer 1**
* * *
General comments

*This paper (acp-2021-239-manuscript-version2) presents a coupled modelling system facilitating a better representation of the processes at the air–sea interface. The system consists of the AROME atmospheric mode, the NEMO ocean circulation model, and the WaveWatchIII ocean wave model. Coupled ocean–atmosphere–wave simulations were performed for a heavy precipitation event (HPE) that occurred between 12 and 14 October 2016 in the South of France, comparing atmosphere-only, coupled atmosphere–wave and ocean–atmosphere simulations. The results are interesting showing that the HPE fine-scale forecast is sensitive to both couplings. The atmosphere-ocean coupling leads to significant changes in the heat and moisture supply of the HPE intensifying convection. The atmosphere-wave coupling mainly leads to changes in the low-level dynamics, affecting the location of the convergence that triggers convection over sea. It is a very good work carefully written with interesting findings supporting the global research trend regarding multi-model coupled systems. So, I can suggest it for publication after some minor revisions and technical corrections.*

Specific Comments

*Lines 236-238, 246-248, 266-276, Table 2 : I have a concern about the methodological design. In studies revealing physical interactions, it is important to change few parameters in each numerical experiment isolating the processes studied. This better supports the scientific reasoning of the study and advantages of coupling can be highlighted. For example, if the initialization of the numerical experiments is based on different data, it is very difficult to explain the reasons of result differences. They may be partially explained by the different initialization and partially by the coupling processes. It is important to note that the*

*aim of the implementation of atmosphere-wave-ocean coupled systems is the "online" physical enrichment of simulations and not the investigation of initialization effects. Sensitivity experiments on initialization is feasible using even uncoupled systems. Lines 266-276 : I appreciate the discussion of Lines 266-276, however, as regards the atmospheric component, did you try to initialize AOW and AO runs using SST analysis instead of the NEMO 7-day SST forecast (spinup)? Also, did you try to initialize atmosphere-only simulations using the NEMO 7-day SST forecast (spinup)? I am wondering how much the result differences are attributed on atmosphere-wave-ocean interactions and how much on different SST initializations. This has to be clarified because determines the scientific findings of this work. I believe that the numerical experiments are comparable only if they have common initial conditions facilitating a more accurate investigation of atmosphere-wave-ocean interactions. An initial SST difference of e.g. 1˚C may cause differences in surface fluxes, convection, atmospheric circulation etc, and thus, hiding coupling benefits. Overall, my suggestion is to perform some additional numerical experiments using either SST analysis or 7-day SST forecast fields, respectively, ensuring that all the experiments are initialized using common SST data. If the new experiments will not result to substantial differences it is not needed to be included in the manuscript.*

3D ocean fields (temperature and salinity ; and not only SST product... ) are mandatory to initiate NEMO and so only ocean 3D products are suitable to initiate any AO/AOW experiments. If PSY4 products (1/12°-resolution) can be used to initiate NEMO-NWMED72, we estimate that the initial choc and adjustment time can perturb the short-range (42h) coupled forecast. This is why we chose to have a 7-day spin-up, that permits the ocean model to adjust to the atmospheric forcing coming from AROME and the development of some fine-scale ocean structures in the near surface layer. In addition, it allows to use instantaneous ocean fields at initial state instead of daily mean fields, as classically available from ocean forecasts or analyses but that are not fully representative of the ocean evolving state when considering short/severe events and/or short-range forecasts. Note that a similar spin-up for the wave model was previously applied (Sauvage et al., 2020).

Indeed, coupling with ocean induces here a change in the SST initial conditions compared to AW. To separate the ocean (wave) coupling effect, the best way is in fact to start with the fully tri-coupled experiments, then decline a numerical set-up with bi-coupled experiments using the same initial forced fields than the tri-coupled one, so the coupling effect could be completely separated from changes in initialisation. This methodology could not be applied at the time we made this study, because in fact we first developed the atmosphere-wave coupling and only then run the oceanic spin-up to add the ocean component. Of course, adding atmosphere-only and atmosphere-waves forecasts using the AOW initial SST appears accurate. Unfortunately, Météo-France recently changed its HPC with an upgrade of the AROME version and adding such experiments would in fact necessitate to redo the whole experimental set.

But, the Mediterranean HPE forecast sensitivity to SST in atmosphere-only simulations as been extensively studied by the past, up to recently (see Meroni et al. 2018, doi :10.1029/2018JD028276 ; Strajnar et al. 2019, doi :10.1002/qj.3425 ; Senatore et al. 2020, doi :10.5194/hess-24-269-2020 for instance). One novelty here is the investigation of more realistic sea surface conditions considering both ocean and waves thanks to coupling and in the air-sea fluxes parameterization, in configuration close to the operational NWP system. We agree that here the ocean coupling includes in fact two effects according to our numerical set-up : a change in the initial SST (with a fine-scale instantaneous condition in AO/AOW over the Mediterranean) and an interactive evolution of the ocean during the forecast. This was fully stand in section 3.2 (lines 266-276) and in Table 2 of the paper. We consider these two both effects as major improvements in the perspective of kilometer-scale short-range forecasts, as discussed in section 5.

Nevertheless, we have modified some parts in the text when possible in order to be the clearest as possible in our results description and discussion (section 4.2), as in the conclusion (section 6) so the impacts of the ocean coupling insertion are more fairly described with "interactive ocean" + "SST initialisation change" considerations.

*Line 102 : A time step of 50 s seems very large to meet CFL criterion in such a high horizontal and vertical resolution. Please check for any typographical error or clarify it.*

AROME has a 50s time-step. This time-step is reachable thanks to the dynamical kernel of AROME (described in Bénard et al. 2010, doi :10.1002/qj.522), which is a two time level, semi-implicit, semi-Lagrangian discretization scheme. Also, the AROME-France dynamics included several changes when building the 1.3 km-resolution configuration (see section 4.1.2 in Brousseau et al. 2016) leading to significant improvements for the time-step.

*Line 111 : Do AROME and SURFEX models use exactly the same grid with 1.3 km horizontal resolution ? I suppose yes (Line 152). What topography, vegetation, land use and soil data are used ? Also, how are you manage land-sea, land-lake and land-river transitions ?*

The horizontal grid is defined in AROME and SURFEX uses the same physical grid. The AROME-France/SURFEX orography is extracted from the Global 30 Arc-Second Elevation Data Set (GTOPO30) database (Gesch et al. 1999, doi :10.1029/99EO00050). Physiographic data are initialized with the ECO-CLIMAP database (Masson et al. 2003, doi :10.1175/1520-0442(2003)16<1261 :AGDOLS>2.0.CO ;2). These two pieces of information have been added in the model description.

As explained in section 2.1.1, each AROME/SURFEX grid box is split into four tiles : land, town, sea, and inland water (lake and river). Transition is ensured as output fluxes are weight-averaged inside each grid box according to the fraction of each respective tile, before being provided to the atmospheric model at every time step. The ocean and wave couplings described here only concerned the sea tile, meaning that only the fluxes computed over the sea tile is sent to OASIS then NEMO, and, the first-level wind

components of any grid box containing a fraction of sea are send to OASIS then WW3.

*Line 151 : Is the 1-hour coupling frequency sufficient for such a high resolution coupled system (1.3 km horizontal resolution)? Have you tried more frequent coupling?*
In Sauvage et al. 2020 [https ://doi.org/10.5194/acp-20-1675-2020], the coupling frequency between atmosphere and waves was set to 1 hour in order to be consistent with the forced mode that used 1-hr forcing fields (*i.e.* the standard AROME output frequency). This 1-hr coupling frequency was found suitable as Figure 2 in Sauvage et al. 2020 and Figure 9 in the present paper show a good agreement of the temporal evolution of both low-level wind and sea state compared to the available buoys. We choose to apply the same frequency for the ocean-atmosphere coupling for consistency and also following the results of Lebeaupin Brossier et al. 2009 [https ://doi.org/10.1007/s10236-009-0198-1] that stated that to well capture the ocean mixed layer response during Mediterranean HPE, a temporal resolution of 1 hour can be sufficient. We did not test any higher coupling frequency. We agree that higher coupling frequency (the minimum frequency with this configuration is 10 min corresponding to 5 NEMO time-steps, 10 WW3 time-steps and 12 AROME time-steps) would possibly have introduced more variability in the results and that is something that should be tested in the future.

*Line 174 : Equation 4 mostly results to negative wind stress values as wind values are almost always larger than surface current ones. Also, Figure 5a shows positive wind stress values. Also, in Equation 1 of your previous study Sauvage et al. (2020) the wind stress is always positive. Please clarify. What is the relation between wind stress and friction velocity as regards the surface layer represented by AROME?*
Equation 4 is a vectorial equation (vectors appear in bold in ACP standard), meaning that it takes into account the two components of both current and wind. Figure 5a shows the wind stress norm $\tau = ||\vec{\tau}|| = \sqrt{\tau_u^2 + \tau_v^2}$.
In Sauvage et al. (2020), the surface current is not considered in fact. Its Equation 1 for stress was not written as a vectorial equation but directly with norms : $\tau = ||\vec{\tau}|| = \rho C_D \Delta U^2 = \rho C_D ||\vec{U}||^2 = \rho u_*^2$, where $u_*$ is the friction velocity. We are sorry it brings confusion. The most adequate equation is Equation 4 here.

*Line 194-195 : Which is exactly the Charnock coefficient formula used in this study?*
The Charnock coefficient formula used here is $\alpha_{ch} = A\chi^{-B}$. In the WASP parametrization the Charnock parameter is piecewise continuously with the coefficients A and B being polynomial functions of the surface wind speed, please see Appendix A of Sauvage et al. 2020 for a detailed description of the Charnock parameter in WASP. The manuscript now refers to Eq. 8 and Appendix A in Sauvage et al. (2020).

*Equation 10 in combination with Equation 9 of Sauvage et al. (2020) presents deep water approxima-*

*tion for wave modelling. Shallow water approximation could be more suitable especially for the coastal areas studied in this work. Please provide explanations for your choice.*

As the Mediterranean Sea is a fetch-limited region the wavelength will almost never exceed twice the water depth which is the criterion to go from deep water to shallow water equation. Thus the phase speed of the waves won't be dependant of the water depth and that is why the deep water equation is the more suitable. In order to better assess that we calculated the minimum wave age $\chi_0$ and peak period $T_0$ for which the deep water formulation would be violated such as $L > 2D$ with $L$ the wavelength and $D$ the depth. Figure A shows the difference between our simulated wave age defined as $\chi = c_p/U_a$ (Eq. 10) and the minimum wave age $\chi_0$ at 12 :00 UTC the 13 October. We can see that the simulated wave age almost never exceed $\chi_0$ meaning that the deep water equation is still valid. Indeed, as the bathymetry becomes quickly deep offshore, in order to exceed $\chi_0$ and for the waves to feel the bottom of the ocean we would have to generate wave peak period up to 60 s, which never happen in this region and explain the large difference obtained in Figure A. It is true that on some points really close to the coast and where the bathymetry is under 200 m like in the Golf of Lion, the shallow water equation might have been used. However, this represents a small area and does not affect our event which is mainly controlled by southerly and easterly winds.

[Figure]

FIGURE A – Wave-age in AW for 13 Oct. 2016 12 :00 UTC compared to the reference wave-age computed with the deep water approximation, and bathymetry (black contours at 0, 200, 1000 and 2000 m) over NWMED.

*Table 6 : Why did you use different number of processes for NEMO (i.e. 8 and 16) in the numerical experiments ? Is this affect the results ?*

As set in italics in Table 6, and detailed lines 470-480, AYSSTatl and AW use a "toymodel" (and not NEMO) only to send the initial SST field trough OASIS. This "toymodel" needs very few computing resources but the AROME-France simulation environment (Olive/Vortex) dictates a minimum value of 8 processes for it.
When NEMO is coupled, it always uses 16 processes.

*Lines 470-493 and Table 6 : I recommend to move technical details and Table 6 in the Appendix.*
This has been modified in the manuscript and added in the Appendix.

*Sections 4, 5, and 6 : The presentation of result differences and the explanations are very good, however, is not clear to me which numerical experiment is the best as regards the overall statistical evaluation using observational data. I am not sure that AOW is the best as expected due to the better physical representation. Please clarify it.*

Although couplings allow more realism, our effort to compare to observations (Figs 7, 9, 11 and Tabs. 3, 4, 5) does not allow to clearly conclude on the forecast improvement (or deterioration) with couplings, mainly because of a general lack of [atmospheric] observations over sea. As stated in the Conclusion further studies need to be conducted over a larger number of cases covering a larger range of weather situations.

**Technical corrections**

Lines 4-7 : The sentence "In order. . . simulations" could be split to two sentences. ok
Lines 5-6 : Be careful with the date format used by the journal. It can be written as "12 and 14 October 2016" similarly to Line 87 of the manuscript. ok : date format has been checked.
Lines 10-14 : The sentence "Even if. . . forecasts" has to be split to two sentences. ok
Lines 21-22 : ECMWF had been adopted atmosphere-wave-ocean full coupling approach before 2019. https ://www.ecmwf.int/en/research/modelling-and-prediction/marine
Yes, indeed, ECMWF includes a wave coupling since 1998 for medium-range forecasts with the coupled IFS-WAM system, but more recently the ocean coupling. In the link you refer to, the ocean coupling description mixes pieces of information for various ECMWF's prediction systems, notably the seasonal/monthly prediction system where NEMO is included since 2013, and the medium-range high-resolution deterministic forecast system where ocean coupling is included operationally since June 2018 from our knowledge (Magnusson et al. 2019 ; see also the first sentence of Browne et al. 2019, ECMWF Technical Memorandum #836).

The first sentence of our introduction (lines 16-22) aims to only refer to short- and medium-range operational NWP systems.

Line 66 : "over a warm" - "over warm" ok

Line 68 : Please clarify "WMED" ok

Line 70 : "re-assert" - "reasserted" ok

Lines 88-91 : I suggest to write Section 2, Section 3 etc. instead of Sect. 2, Sect. 3 etc. ok

Line 95 : "models configurations" - "model configurations" ok

Line 121 : "has 2 two open" - "has two open" ok

Line 132 : "consists in" - "consists of" ok

Line 142 : Maybe "adopted" ? ok : "is adapted from" has been replaced by "is from" ...

Lines 177-178 : Please better clarify $\Delta\theta$ and $\Delta q$ of Equation 5. E.g., do you mean $(\theta s - \theta a)$ for $\Delta\theta$?

Yes indeed we meant $(\theta s - \theta a)$ and $(qs - qa)$. This has been clarified in the text.

Line 281 : "couplings" - "coupling" ok

Table 1 : "meridian" - "meridional" ok
* * *
**Reviewer 2**
* * *
*This study assesses the impact of the coupling between ocean and atmopshere, atmoshere and waves, and the tri-coupling atmosphere-ocean-wave (without taking into account the impact of waves on ocean characteristics) on a heavy Precipitation event which occured over France and Mediterranean Sea using the forecast system developped at Meteo-france. It consists of the AROME model for atmosphere, Nemo for ocean and Wave-watchIII for waves. The paper is well organised and shows in a convincing way that the coupling between atmosphere and ocean affects the heat and moisture supply to the atmosphere and hence the accumulated precipitation in the convective system, while the coupling between atmosphere and waves modifies the wind stress and wind speed, impacting the location of the system in the forecast. A discussion on the numerical impact of each coupling is added at the end of the manuscript helping the reader to have an idea of the benefit/loss of each coupling. I think this work is a very nice contribution to the scientific community in its effort to improve the forecast of High impact Weather event, and thus deserves to be published after minor modifications needed to clarify some results.*

Main comments

*- the WASP parametrization is described in details in the document and the equations reveal the strong dependence of several terms to the wind at the first level of the atmospheric model : in this forecast system, the lower level is at about 5m. Would it be possible to have more details on the vertical distribution of wind in the model, and a better assessment of the sensitivity of the results to the height of the first model level ?*
This is a good remark from the reviewer which actually go beyond the scope of this study. As we wanted here to place ourselves in a forecast point of view, we did not test the sensitivity to different wind levels and used the vertical resolution of AROME. Also, the coupling is done in SURFEX were there is no vertical profiles. More generally this problematic addresses how typically the sea surface fluxes bulk algorithms relate the ocean surface fields to the near-surface atmospheric fields in the frame of Monin-Obukhov similarity theory. Recently Pelletier et al. 2021 [DOI : 10.1002/qj.3991] discussed a new approach introducing a more continuous profiles between the ocean surface and near atmospheric fields.

*- following the same idea of better assessing the sensitivity of the results to the choice of the model configuration : how much the frequency of coupling can affect the results ?*
Indeed this point have also been raised by the Reviewer #1. Although we did not test any other coupling frequency we agree that higher coupling frequency would possibly have introduced more variability in the results and that is something that should be tested in the future. However, Figure 9 in the present paper and Figure 2 in Sauvage et al. 2020 [https ://doi.org/10.5194/acp-20-1675-2020] showed a good agreement of the temporal evolution of both low-level wind and sea state compared to the available buoys using 1h coupling during this event. Moreover, Lebeaupin et al. 2009 [https ://doi.org/10.1007/s10236-009-0198-1] stated that for the ocean, using a finer resolution than 1h for the atmospheric forcing was not necessary during Mediterranean HPE. They also showed that using coarser temporal resolution (3h or 6h) could induced strong bias in the fine-scale ocean mixed layer response.

*- isn't it a problem in the interpretation of the results the fact that the initial state of SST is different when the atmosphere is coupled to the ocean or not ?*
This is a good remark also raised by the Reviewer #1, please see pages 2 and 3. The ocean coupling indeed includes in fact two effects according to our numerical set-up : a change in the initial SST over the Mediterranean (with a fine-scale instantaneous condition in AO/AOW) and an interactive ocean during the forecast. This was fully stand in section 3.2 (lines 266-276) and in Table 2 of the paper. We consider these two both effects as major improvements in the perspective of kilometer-scale short-range forecasts, as discussed in the Section 5 of the manuscript.

Minor comments

*- section 4.1 : why is the reduction of latent and sensible heat fluxes stronger in AOW compared to AO than between AW and AY ?*

In AO and AOW, the atmospheric low levels are warmer and moister in the Azur area during phases I and II. In this more unstable environment, the enthalpy is higher following the Clausius-Clapeyron relationship, which implies that the turbulent heat fluxes do respond not linearly depending on stability. For this reason, the wave impact results in a slightly higher decrease in turbulent heat fluxes comparing AOW/AO than when comparing AW/AY. This explanation has been briefly added to the concerned sentence and the value of the decrease when comparing AOW/AO has been precised to 3% in section 4.1.

*- section 4.1, l320 : the authors say that this reduction is mainly due to the slow down of the wind but did they check the others parameters ?*

In our configuration where WW3 is not directly coupled with NEMO we did not see any significant impact on the SST due to the coupling of waves with the atmosphere. Moreover, the impact of waves on the roughness length is limited to the drag coefficient $C_D$ and does not affect $C_E$ and $C_H$, as verified in Figure B.

*- section 4.2 : if more moisture is extracted from the ocean to the atmospheric low level in AOW than in AW as said on l.360, why don't we observe a modification of RH2m (l.373) ?*

Relative humidity was chosen to be directly compared to observed values. However, it depends on air temperature, in the way that with an increase in air temperature, relative humidity decreases for the same water vapour content. Figure C shows the simulated specific humidity at 2m, with the comparison between AOW and AW. It highlights the increase in humidity at low-level with the insertion of the ocean, in particular during the two first phases of the event and in the Azur area (around +5-10%). This increase is consistent with larger latent heat fluxes induced by a higher initial SST in this area (see Figure 3c in the paper). Differences are then less important with time in the upstream zone, because of the ocean mixed layer cooling induced by mixing and surface evaporation/moisture extraction. In addition, there is a significant role of horizontal advection, with moisture transport from the extraction area toward the convective systems.
The increase in 2m-specific humidity is now mentioned in the text.

*- Fig.10 : light blue lines for reflectivity are very difficult to see.*
Reflectivities are now drawn in yellow in Figure 6,b,d,f to be more readable.

[Figure]

FIGURE B – Roughness length ($z_0$), drag coefficient ($C_D$), and transfer coefficient for moisture ($C_E$) depending on SST (°C) or 10m-wind speed (m s$^{-1}$), in the Azur area for range +24h of the five forecasts (AOW is repeated) starting on 00 UT 13 October 2016

[Figure]

FIGURE C – Specific humidity at 2 meter-height (kg/kg) in AW (top panels), absolute (kg/kg, middle panels) and relative (%, bottom panels) differences between AOW and AW, at +12 (left), +14 (center) and +24h (right) of the forecasts starting on 00 UT 13 October 2016.

References :

Bénard, P., Vivoda, J., Mašek, J., Smolíková, P., Yessad, K., Smith, C., Brožková, R., Geleyn, J.-F. : Dynamical kernel of the Aladin–NH spectral limited-area model : Revised formulation and sensitivity experiments. Q.J.R. Meteorol. Soc., 136 : 155-169. https ://doi.org/10.1002/qj.522, 2010.

Brousseau, P., Seity, Y., Ricard, D., Léger, J. : Improvement of the forecast of convective activity from the AROME-France system, Quart. J. Roy. Meteor. Soc., 142, 2231–2243, https ://doi.org/10.1002/qj.2822, 2016.

Browne, P., de Rosnay, P, Zuo H., Bennett, A., Dawson A. : Weakly coupled ocean-atmosphere data assimilation in the ECMWF NWP system. ECMWF Technical Memorandum #836, https ://www.ecmwf.int/node/18814, https ://doi.org/10.21957/eqe8rx02, 2019.

Gesch, D. B., Verdin, K. L., and Greenlee, S. K. : New land surface digital elevation model covers the Earth, Eos Trans. AGU, 80(6), 69– 70, https ://doi.org/10.1029/99EO00050, 1999.

Lebeaupin Brossier, C., Ducrocq, V., Giordani, H. : Effects of the air–sea coupling time frequency on the ocean response during Mediterranean intense events. Ocean Dynamics 59, 539–549, https ://doi.org/10.1007/s10236-009-0198-1, 2009.

Magnusson, L., Bidlot, J.-R., Bonavita, M., Brown, A., Browne, P., De Chiara, G., Dahoui, M., Lang, S., McNally, T., Mogensen, K., et al. : ECMWF activities for improved hurricane forecasts, Bulletin of the American Meteorological Society, 100, 445–458, https ://doi.org/10.1175/BAMS-D-18-0044.1, 2019.

Masson, V., Champeaux, J., Chauvin, F., Meriguet, C., Lacaze, R. : A Global Database of Land Surface Parameters at 1-km Resolution in Meteorological and Climate Models, Journal of Climate, 16(9), 1261-1282, https ://doi.org/10.1175/1520-0442(2003)16<1261 :AGDOLS>2.0.CO ;2, 2003.

Meroni, A. N., Parodi, A., Pasquero, C. : Role of SST patterns on surface wind modulation of a heavy midlatitude precipitation event. Journal of Geophysical Research : Atmospheres, 123, 9081– 9096. https ://doi.org/10.1029/2018JD028276, 2018.

Pelletier, C, Lemarié, F, Blayo, E, Bouin, M-N, Redelsperger, J-L. : Two-sided turbulent surface-layer parameterizations for computing air–sea fluxes. Q. J. R. Meteorol. Soc., 1726– 1751. https ://doi.org/10.1002/qj.3991, 2021.

Sauvage, C., Lebeaupin Brossier, C., Bouin, M.-N., Ducrocq, V. : Characterization of the air–sea exchange mechanisms during a Mediterranean heavy precipitation event using realistic sea state modelling, Atmos. Chem. Phys., 20, 1675–1699, https ://doi.org/10.5194/acp-20-1675-2020, 2020.

Senatore, A., Furnari, L., and Mendicino, G. : Impact of high-resolution sea surface temperature representation on the forecast of small Mediterranean catchments' hydrological responses to heavy precipitation, Hydrol. Earth Syst. Sci., 24, 269–291, https ://doi.org/10.5194/hess-24-269-2020, 2020.

Strajnar, B, Cedilnik, J, Fettich, A, et al. : Impact of two-way coupling and sea-surface temperature on precipitation forecasts in regional atmosphere and ocean models. Q. J. R. Meteorol. Soc. ; 145 : 228– 242. https ://doi.org/10.1002/qj.3425, 2019.